# Exploring Efficient Few-shot Adaptation for Vision Transformers

**Chengming Xu**                                                                  *cmxu18@fudan.edu.cn*
*School of Data Science, Fudan University*

**Siqian Yang**                                                                  *seasonsyang@tencent.com*
**Yabiao Wang**                                                                  *caseywang@tencent.com*
*Youtu Lab, Tencent*

**Zhanxiong Wang**                                                               *maxzxwang@tencent.com*
*Tencent*

**Yanwei Fu**[*]                                                                  *yanweifu@fudan.edu.cn*
**Xiangyang Xue**                                                                *xiangyangxue@fudan.edu.cn*
*School of Data Science, Fudan University*

**Reviewed on OpenReview:** *https://openreview.net/forum?id=n3qLz4eL1l*

## Abstract

The task of Few-shot Learning (FSL) aims to do the inference on novel categories containing only few labeled examples, with the help of knowledge learned from base categories containing abundant labeled training samples. While there are numerous works into FSL task, Vision Transformers (ViTs) have rarely been taken as the backbone to FSL with few trials (Hu et al., 2022; Evci et al., 2022; Abnar et al.) focusing on naïve finetuning of whole backbone or classification layer. Essentially, despite ViTs have been shown to enjoy comparable or even better performance on other vision tasks, it is still very nontrivial to efficiently finetune the ViTs in real-world FSL scenarios. To this end, we propose a novel efficient Transformer Tuning (eTT) method that facilitates finetuning ViTs in the FSL tasks. The key novelties come from the newly presented Attentive Prefix Tuning (APT) and Domain Residual Adapter (DRA) for the task and backbone tuning, individually. Specifically, in APT, the prefix is projected to new key and value pairs that are attached to each self-attention layer to provide the model with task-specific information. Moreover, we design the DRA in the form of learnable offset vectors to handle the potential domain gaps between base and novel data. To ensure the APT would not deviate from the initial task-specific information much, we further propose a novel prototypical regularization, which minimizes the similarity between the projected distribution of prefix and initial prototypes, regularizing the update procedure. Our method receives outstanding performance on the challenging Meta-Dataset. We conduct extensive experiments to show the efficacy of our model. Our code is available at https://github.com/loadder/eTT_TMLR2022.

## 1 Introduction

Modern computer vision models such as ResNet (He et al., 2016) and Faster R-CNN (Ren et al., 2015) are trained on large-scale training sets, and not well generalize to handle the long tail categories with few

---

[*]This paper is supported by the project NSFC(62076067).

labeled samples. Few-shot Learning (FSL) has thus been studied to make inference on insufficiently-labeled *novel* categories typically with the transferable knowledge learned from *base* categories which are provided with abundant labeled training samples. Essentially, the FSL can be taken as *representation learning*, as its backbones should ideally extract features representative and generalizable to various novel tasks. Currently Convolutional Neural Networks (CNNs), especially ResNet, are the predominant backbone and widely utilized in most existing FSL works (Ravi & Larochelle, 2017; Finn et al., 2017; Nichol et al., 2018; Li et al., 2017; Sun et al., 2019).

Recently, by taking the merits of Multi-headed Self-Attention (MSA) mechanism and Feed Forward Network (FFN), the transformers have been widely used in the recognition (Alexey et al.; Liu et al., 2021b), detection (Beal et al., 2020) and image editing (Cao et al., 2021). The general pipeline of Pretrain-(Meta-train)-Finetune has been explored in few ViTs on FSL (Hu et al., 2022; Evci et al., 2022; Abnar et al.), recently. Particularly, the ViT models are first pretrained/meta-trained on a large-scale dataset. Then a test-time finetune procedure is set up for each target task on novel data. The finetuning strategy can be generally categorized into linear classifier probing and backbone tuning: the former one optimizes the reasonable decision boundaries by the fixed embeddings, while the latter one considers the adaptation of both embedding space and classifier.

In this paper we focus on the backbone tuning method. (Hu et al., 2022) shows that the naïve Pretrain-Meta-train-Finetune (P>M>F) baseline can generally have satisfactory performance in FSL. Unfortunately, it involves heavy computations and potential overfitting in FSL setting. Particularly, (1) It typically demands extraordinary computing power to formulate episodes from a large number of support classes to update the whole network parameters. Thus it is less efficient in many real-case applications. For example, the edge devices such as mobiles donot have enough computational power to adapt all model parameters by personalized/specialized data collected on these devices. (2) It is very subtle and difficult to directly fine-tune trained deep models on one or two labeled instances per class, as such few-shot models will suffer from severe overfitting (Snell et al., 2017; Fei-Fei et al., 2006; Brian et al.). By contrast, humans have the ability of conducting few-shot recognition from even single example of unseen novel category with very high confidence.

Such problems may be the culprit of the phenomenon that their proposed finetune strategy only works on part of datasets and has less effect to the others. This suggests their limited usage of ViT backbone for any potential FSL applications. An alternative choice is to finetune specific layers in a ViT model with much smaller tunable parameters (ViT-s block in Fig. 1(a)). Such a strategy nevertheless can only finetune either low-level or high-level features, leading to inferior performance in many cases. Therefore it is desirable to have an efficient and light-weighted ViT tuning method that shall not only avoid overfitting to small training samples, but also achieve high performance of FSL.

In this paper, we present a novel efficient Transformer Tuning (eTT) for few-shot learning task, which adopts a pretrain-finetune pipeline. To pretrain our transformer, we advocate utilizing the recent self-supervised method – DINO (Caron et al., 2021). Our key novelties are in the finetuning stage. As illustrated in Fig. 1(b), we propose Attentive Prefix Tuning (APT) and Domain Residual Adapter (DRA) as the key components to our eTT, to efficiently learn the newly-introduced tunable parameters over novel support sets. Specifically, we formulate the attentive prototypes by aggregating patch embeddings with the corresponding attention weights of the class token for each image, so as to provide the model with abundant task-specific information and guide each self-attention layer to aggregate more class-related features. To encourage the prefix to keep the prior knowledge from initial prototypes, we further propose a novel prototypical regularization which restricts the relationship between the prefix and prototypes by optimizing the similarity of their projected distributions. Moreover, we propose to additionally adopt a light-weighted domain residual adapter in the form of learnable offset to deal with the potential failure of APT on large domain gaps. Extensive experiments are conducted to evaluate our eTT: we use the ViT-tiny and ViT-small backbones on the large-scale Meta-Dataset (Triantafillou et al., 2019) consisting of ten sub-datasets from different domains; and the results show that our model can achieve outstanding performance with comparable or even much fewer model parameters. Thus our eTT is a promising method on efficiently finetuning ViTs on the FSL tasks.

Our paper has the following contributions.
1. In order to solve the problem of inefficiency and make better use of ViT in FSL, we propose a novel

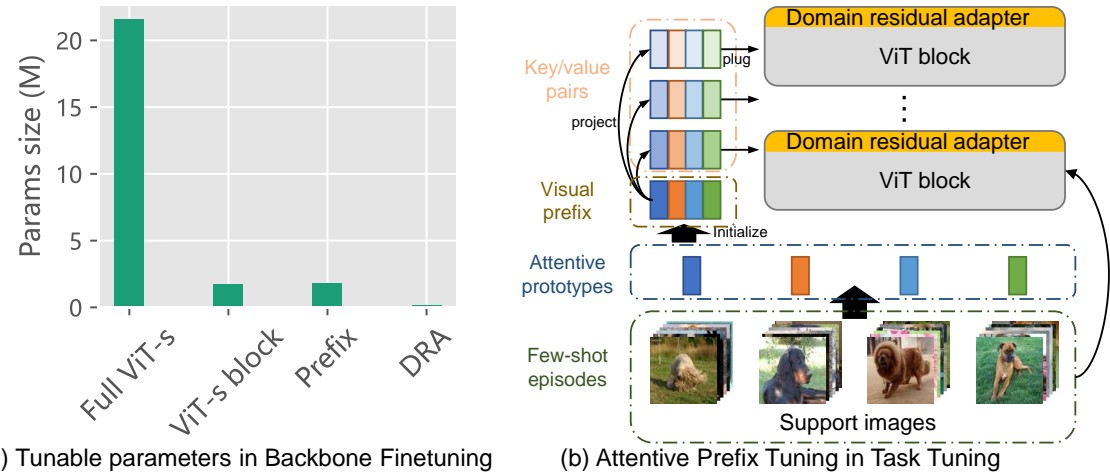

(a) Tunable parameters in Backbone Finetuning     (b) Attentive Prefix Tuning in Task Tuning

Figure 1: (a) Comparing with other backbones, we propose the Domain Residual Adapter (DRA) to tune much less parameters in our efficient Transformer Tuning (eTT); and effective for large-scale FSL. (b) The few-shot support samples are first processed into attentive prototypes which are used to initialize the task-specific visual prefix. Then the prefix together with the domain adapter are attached to each layer of the ViT to finetune our ViTs.

finetuning method named efficient Transformer Tuning (eTT).

2. Inspired by recent advance in language model, a novel attentive prefix tuning is presented utilizing the attentive prototypes to embed the task-specific knowledge into pretrained ViT model. Particularly, we propose a new initialization strategy tailored for FSL by leveraging prototypical information from the self-attention layers. Moreover, a novel domain residual adapter is repurposed to handle the various domain gaps between training and testing data.

3. We introduce a prototypical regularization term which can constrain the update procedure of prefix during finetuning to maintain the initial task-specific knowledge.

4. By utilizing the proposed eTT, our ViT models receive remarkable performance on Meta-Dataset, overpassing the existing ResNet-based methods without using additional training data. More importantly, both of the model scale and efficiency of our method are comparable with the other competitors, indicating the promising application of ViTs in FSL.

## 2 Related Works

**Few-shot recognition.** FSL learns transferable knowledge from base classes and adapt it to a disjoint set (novel classes) with limited training data. Among those FSL tasks, few-shot image recognition is the one with most focus and researches. Existing works can be grouped into two main categories. One is optimization-based methods (Ravi & Larochelle, 2017; Finn et al., 2017; Nichol et al., 2018; Li et al., 2017; Sun et al., 2019), which learn parameters that can be better finetuned on few-shot support sets. The other is metric-based methods such as ProtoNet (Snell et al., 2017), RelationNet (Sung et al., 2018), CAN (Hou et al., 2019), DMF (Xu et al., 2021), COSOC (Luo et al., 2021) and CTX (Doersch et al., 2020), which solve FSL by applying an existing or learned metric on the extracted features of images. Particularly, CTX (Doersch et al., 2020) builds up a cross attention module which interacts between query and support images to adaptively aggregate better prototypes than simply averaging all support features. While these methods perform well on classical few-shot learning settings, most of them adopt convnet as backbone, especially ResNet (He et al., 2016). We, on the opposite, try to make full use of another widely-applied structure, i.e. ViT, in FSL, which requires extra design for training and finetuning strategy.

**Transformer in vision tasks.** Transformers widely utilize the self-attention mechanism which originally are employed to process the feature sequence in Vaswani et al. (2017). Then large scale transformers become increasingly popular in NLP tasks to build complex language models, and also extend to vision tasks (Alexey

et al.; Yuan et al., 2021; Liu et al., 2021b) by formulating the token sequence with image patches processed with position embedding. It has been shown the efficacy in various applications, such as (Liu et al., 2021a) for image caption, (Sun et al., 2020) for multiple object tracking and (Esser et al., 2021; Cao et al., 2021) for image inpainting and editing. Critically, ViTs is typically trained by very large-scale dataset, and few effort has been dedicated in training or finetuning on few-shot supervision. We follow the pretrain-meta-train-finetune pipeline (Hu et al., 2022), while their method finetune the whole ViTs on few-shot examples, and thus has less efficiency and can easily overfit. In contrast, our proposed eTT has the key components of DRA and APT, demanding much less tunable parameters with much better performance.

**Finetuning algorithm for ViT.** The idea of finetuning ViTs on small-scale datasets has been partly investigated in Natural Language Processing (NLP) communities. Houlsby et al. (2019) proposed to attach two learnable bottleneck adapters to each transformer layer. Other works (Xiang & Percy; Brian et al.) make use of the prompt which trains a small task-specific prompt for each task so that the prompt can guide the model with knowledge corresponding to the task. Such a prompting idea from NLP is inherited and repurposed to finetune a learnable prefix for each novel episode in this paper. However, these works (Xiang & Percy; Brian et al.; Houlsby et al., 2019) initialize the prefix or prompt with word embeddings which is not available in our problem. Instead, we propose an attentive prototype with regularization initializing the visual prefix with object-centric embeddings. Additionally, we notice that a very good concurrent technical report (Jia et al., 2022) also studies finetuning visual prompt for pretrained ViTs in downstream tasks. We highlight the two key differences from our eTT. The first is about the initialization. While initialization strategy does not matter in their method and the corresponding tasks, we show in our experiments that randomly initializing prefix does lead to sub-optimal performance in FSL, which leads to the necessity of a well-designed initialization. The second is that we further propose a regularization term to restrict the prefix, which has never been studied in existing works.

**Task-specific Adapter.** The idea of task-specific adapter has been explored in several works like (Li et al., 2022; Rebuffi et al., 2017) to adapt CNNs to learn the whole information from support set. Besides, (Requeima et al., 2019; Bateni et al., 2020) adopt Feature-wise Linear Modulation (FiLM) layers (Perez et al., 2018) to adapt task-specific information into networks. In contrast, we repurpose the adapter as the domain residual to update transformer blocks in a more light-weighted way with less learnable parameters. Beyond different structures, our proposed DRA intrinsically serves as the domain adapter rather than meta-learner for the FSL in Rusu et al. (2018); Sun et al. (2019); Li et al. (2022); Requeima et al. (2019). While these previous works require meta-training to optimize their adaptation modules, our method simply utilizes the novel support data to learn the DRA, thus reducing the training cost. Furthermore, our DRA is mostly tuned to bridge the visual domain gap between base and novel categories, thus improving the learning of APT on each episode task.

## 3 Methodology

### 3.1 Problem Setup

We formulate few-shot learning in the meta-learning paradigm. In general, we have two sets of data, namely meta-train set $\mathcal{D}_s = \{(\mathbf{I}_i, y_i), y_i \in \mathcal{C}_s\}$ and meta-test set $\mathcal{D}_t = \{(\mathbf{I}_i, y_i), y_i \in \mathcal{C}_t\}$ which contain the base and novel data respectively and are possibly collected from different domains. $\mathcal{C}_s$ and $\mathcal{C}_t$ ($\mathcal{C}_s \cap \mathcal{C}_t = \emptyset$) denote base and novel category sets. FSL aims to train a model on $\mathcal{D}_s$ which is generalizable enough on $\mathcal{D}_t$. In the testing phase, the model can learn from few labelled data from each category of $\mathcal{C}_t$.

While most previous FSL works (Snell et al., 2017; Sung et al., 2018) utilize the setting of $N$-way $K$-shot in mini-ImageNet, i.e., $K$ training samples from $N$ class, we follow CTX (Doersch et al., 2020) to adopt the setting on the large-scale Meta-Dataset (Triantafillou et al., 2019). In each episode $\mathcal{T}$, $N$ is first uniformly sampled from $[5, N_{max}]$ where $N_{max}$ equals to $\min(50, |\mathcal{C}_t|)$ or $\min(50, |\mathcal{C}_s|)$ on training or testing stage, accordingly. $N$ is supposed to be accessible knowledge during both training and testing. In the most naïve case, one can get $N$ by directly counting the number of support classes. From each of the sampled category, $M$ query samples per category are randomly selected, and thus constructing the query set $\mathcal{Q} = \{(\mathbf{I}_i^q, y_i^q)\}_{i=1}^{N_Q}$. After that random amount of samples are taken from the rest of samples belonging to these categories to

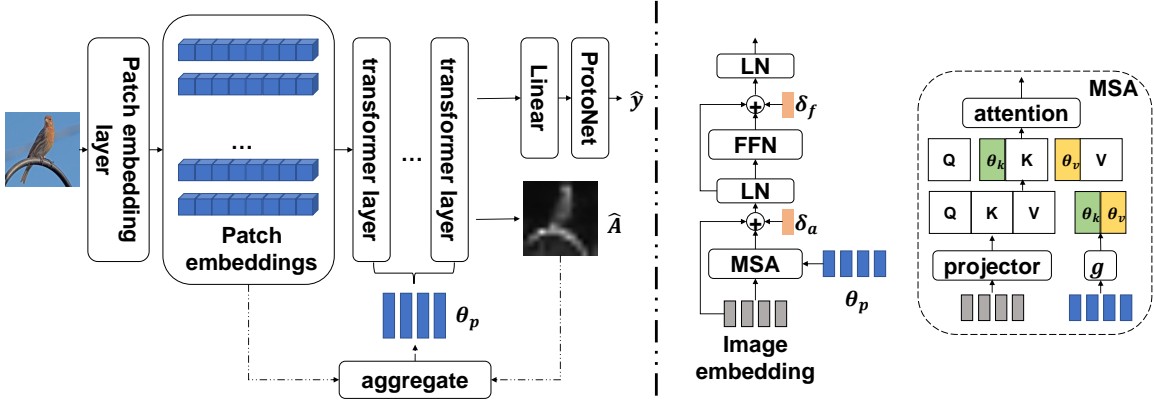

Figure 2: Schematic illustration of our proposed model. For each image, we first fetch its patch embedding sequence and the attention score with regard to the last layer's class token, from which the image embedding can be computed. Then the visual prefix is initialized as the attentive prototypes of image embeddings. The prefix, together with the proposed domain residual adapter are attached to the model. The final features are processed with an extra linear transformation layer and predicted with ProtoNet. Dashed arrows denote forward propagation before test-time finetuning.

form the support set $\mathcal{S} = \{(\mathbf{I}_i^{\mathrm{supp}}, y_i^{\mathrm{supp}})\}_{i=1}^{N_S}$. Note that compared to the classical $N$-way $K$-shot setting, such a setting generates class-imbalanced support sets, and different episodes contain different numbers of support samples. This is much more challenging to the model and learning algorithms, as they shall handle both extremely large and small support sets.

### 3.2 Overview of Our Method

To handle the optimization of various episodes on large-scale dataset, we present our novel finetuning model – efficient Transformer Tuning (eTT) as shown in Fig. 2. Our eTT follows the pipeline in Hu et al. (2022), and has key stages of the pretraining and finetuning. We employ DINO as pretraining, and conduct the task tuning by attentive prefix tuning (Sec. 3.4), and backbone tuning with domain residual adapter (Sec. 3.5).

**Pre-training**. As previous work (Hu et al., 2022) shows the importance of self-supervised pre-training to learning vision transformer models, we adopt the same principle and introduce the self-supervised learning model to pre-train our ViT backbone on base data. Specifically, we utilize the recent State-of-the-art self-supervised ViT models – DINO (Caron et al., 2021) to pretrain our model. DINO builds up supervision based on a self-distillation framework by using the multi-crop strategy (Caron et al., 2020). As we will show in our experiments, such a pre-trained model shall have good cluster property even among cross domain images, potentially benefiting our following FSL stages. Note that different from (Hu et al., 2022) which takes an off-the-shelf model pretrained with DINO on full ImageNet, we strictly follow the FSL protocols to retrain the DINO models on the meta-train split in the target dataset to avoid the abuse of extra data.

One would ask whether it is necessary to make use of the annotations for base data, since supervised pretrain has been proven to be effective in many previous FSL works (Ye et al., 2020; Hou et al., 2019). As we will show in the experiments, an additional finetuning with image labels on base data cannot bring consistent improvement and even makes it worse on most datasets, which may be caused by the overfitting on the image labels leads to less generalization ability across different domains. Moreover, compared with vanilla supervised training, the attention maps for models trained by DINO contain more semantic information, which we will utilize in the following context.

### 3.3 Preliminary: Vanilla Test-time Finetuning

Before fully developing our fine-tuning contributions, we review the simple and effective finetuning method named LT+NCC (Li et al., 2021). The novel modules proposed by us in the following context are all adopted together with this simple baseline method. Given a ViT backbone $f_\theta$ that is parameterized by $\theta$ and an

episode $\mathcal{T}$, the support features $\{x_i^{supp}\}_{i=1}^{N_S}$, where $x_i^{supp} = f_\theta(\mathbf{I}_i^{supp})$, are extracted from the support set $\{\mathbf{I}_i^{supp}\}_{i=1}^{N_S}$. Then, a learnable linear transformation $\phi$ is added to the model to realize the adaptation, which results in the final support features used for classification $\{\hat{x}_i^{supp}\}_{i=1}^{N_S}$, where $\hat{x}_i^{supp} = \phi(x_i^{supp})$. The prediction of these support images can thus be calculated based on the similarity between the transformed features and the aggregated prototypes as,

$$\bar{x}_c = \frac{1}{\sum_{i=1}^{N_s} \mathbb{1}_{y_i^{supp}=c}} \sum_{i=1}^{N_s} \hat{x}_i^{supp} \mathbb{1}_{y_i^{supp}=c} \qquad \hat{y}_i^{supp}(c) = \frac{\exp(d(\hat{x}_i^{supp}, \bar{x}_c))}{\sum_{c=1}^{N} \exp(d(\hat{x}_i^{supp}, \bar{x}_c))} \tag{1}$$

where $d$ denotes cosine similarity, i.e., $d(a,b) = \frac{a^T b}{\|a\|\|b\|}$. We fix all of the parameters in the original backbone, and adopt the cross entropy loss to optimize the transformation $\phi$. Precisely speaking, for each support image $\mathbf{I}^{supp}$ together with its annotation $y^{supp}$, the objective function is as following:

$$\ell_{CE} = -y^{supp} \cdot \log \hat{y}^{supp} \tag{2}$$

After finetuning, $\phi$ is applied to query features and the same procedure as above is performed between the processed query features $\{\hat{x}_i^q\}$ and prototypes $\{\bar{x}_c\}_{c=1}^{N}$ for the inference of each episode.

### 3.4 Task Tuning by Attentive Prefix Tuning

We finetune the pre-trained ViT with support set via an attentive prefix tuning strategy. Specifically, a prefix matrix $\theta_P \in \mathbb{R}^{N_P \times d}$ is first initialized, where $N_P$ denotes the number of prefix. Then a bottleneck $g$ is added upon $\theta_P$ to produce $\hat{\theta}_P \in \mathbb{R}^{N_P \times (2Ld)}$, where $L$ denotes the number of backbone layers. The $g$ plays the same role as the projector in each self-attention layer, except that all layers share the same module. The produced $\hat{\theta}_P$ can be reshaped and seen as $L$ value and key pairs $\{\theta_v^l, \theta_k^l\}_{l=1}^{L}, \theta_v^l, \theta_k^l \in \mathbb{R}^{N_P \times d}$. The MSA block in the $L$-th layer can then be reformed by attaching these new pairs to the original key and value sequences:

$$A^l = \text{Attn}(Q, [K; \theta_k^l]) \qquad \text{output} = A^l [V; \theta_v^l] \tag{3}$$

where $[\cdot; \cdot]$ denotes concatenation, Attn denotes the calculation of MSA matrices. In this way, the prefix can affect the attention matrix $A^l$ and result in different output features from the original ones.

**Remark**. Compared with the naive strategy that finetunes specific layers in ViT (ViT-s block in Fig. 1(a)) which can only adjust part of blocks, the prefix can evenly adapt each layer's image embedding with almost the same parameter size as one transformer layer, as shown in Tab. 1(a). By fixing the model parameters and optimizing the prefix $\theta_P$ and the transformation module $g$, the support knowledge can be smoothly embedded into the prefix, which further helps the task adaptation.

**Attentive Prototype**. The initialization of the prefix is very important to our APT, as it greatly boosts the performance. Critically, quite different from the prefix or prompt tuning in NLP and visual-context tasks that have task-specific instructions explicitly as word embedding sequences, each episode in our FSL only has the few support images and their labels. Thus, rather than steering the model with *'what should be done'* as in Xiang & Percy, our APT shall provide the model with *'what we have globally'* by leveraging the class-specific information. Thus, the attentive prototype is presented to aggregate the image embeddings with object-centric attention, as the initialization of the prefix. Particularly, each support image $\mathbf{I}^{supp}$ is first transformed to a patch embebdding sequence $\{\tilde{x}_m^{supp}\}_{m=1}^{P}$ with the starting patch embedding layer,

$$\tilde{x}_m^{supp} = f_{\theta_{pe}}(I_m^{supp}) + E_m^{pos} \tag{4}$$

where $m = 1, \cdots, P^2$ is the patch index; $f_{\theta_{pe}}$ denotes the patch embedding layer which is typically a convolutional layer whose kernel size equals to patch size; and $E^{pos}$ indicates the position embedding. Meanwhile, we can get unnormalized attention score $A \in \mathbb{R}^{h \times P}$ between the class token and image patches from the last MSA layer, where $h$ denotes number of heads in each MSA module. Such an attention vector can focus on the foreground in the image, especially for models trained with DINO (Caron et al., 2021), with each head indicating a particular part or an object. We can thus get the initial image-level representation

$$\hat{A} = \sigma(A) \qquad \tilde{x}^{supp} = \frac{1}{h} \sum_{n=1}^{h} \sum_{m=1}^{P^2} \hat{A}_{nm} \tilde{x}_m^{supp} \tag{5}$$

where $\sigma$ is softmax function. Compared with simply averaging all patch embeddings, the attentive embeddings can highlight the objects of interest and suppress the background information. Then the prototypes $\bar{x}$ can be calculated by averaging the attentive image embeddings belonging to each support category. We set the number of prefix as $N$, which is available during testing for each episode, and initialize the prefix with $\bar{x}$.

**Remark**. In this way, commonly-used prototypes can provide the model with comprehensive information about the episode. Also such a first-order statistics is comparable with the normal patch features among the layers. This can benefit the training with more stability. When $N$ is large, more prefix are required to fully learn the information included by each episode. On the other hand, when $N$ is small so that the episode is relatively easy, fewer prefix can handle the support knowledge without trouble while decreasing the computing debt.

### 3.5 Backbone Tuning by Domain Residual Adapter

Finetuning few-shot tasks by APT will make a good balance between performance and efficiency. To further improve the model generalization ability on different domains, we further propose the backbone tuning by leveraging the Domain Residual Adapters (DRA), as illustrated in Fig. 2. Specifically, for the $l$-th transformer layer, we attach two learnable offset vectors $\delta_a^l, \delta_f^l \in \mathbb{R}^d$ to the MSA and FFN. After features are processed with MSA and FFN, the corresponding offsets are added to them so that the extreme domain gap can be neutralized. These offsets are expected to represent the gap between source and target domains, and transfer the original manifold to a more appropriate one.

### 3.6 Loss Functions

**Prototypical Regularization**. In addition to the cross entropy loss in Eq. 2, we propose a novel prototypical regularization to ensure the class-specific information, which is embedded in the prefix via initialization, can be maintained during update. The knowledge in attentive prototypes is distilled to the prefix during finetuning. Concretely, in each iteration, the prototypes $\bar{x}$ and prefix $\theta_P$ are first projected to a latent space via a projector module $\psi$, which produces $\bar{x}'$ and $\theta_P'$ respectively. Then the distillation loss is computed using these two embeddings as,

$$\ell_{dist} = \frac{1}{N} \sum_{n=1}^{N} H(\bar{x}'^n, \theta_P'^n) \tag{6}$$

where $H(a, b) = -a \log b$. The above objective function can ensure the prototype of each category and the corresponding prefix contain consistent information, which is indicated by the similarity between distributions after projection. To make training more stable and avoid collapse, for each episode we maintain an exponential moving average (EMA) of $\bar{x}'$ as the center variable $c_{center}$. Before calculating $\ell_{dist}$, we standardize $\bar{x}'$ as $\sigma(\frac{\bar{x}' - x_{center}}{\tau})$, where $\sigma$ denotes softmax function and $\tau$ is the temperature typically set as 0.04.

Once having both of the above losses calculated, we can optimize the model parameters including the DRA, the prefix together with the transformation $g$ and the projector $\psi$, with the following objective function:

$$\mathcal{L} = \ell_{CE} + \lambda \ell_{dist} \tag{7}$$

where the scalar weight $\lambda$ controls the strength of the regularization.

**Remarks**. For a ViT with $L$ layers, $n_h$ heads and $d$ feature dimension, the size of trainable parameters is $(N + d' + d_{proj} + d)d + 2(d' + 1)Ld$, where $d'$ is the hidden dimension for transformation module $g$ and $d_{proj}$ denotes output dimension for the projector $\psi$, which is much smaller than that of the whole backbone model. Specifically, the learnable modules during finetuning have only about 9% parameters with regard to the whole transformer model when using ViT-small and ViT-tiny.

## 4 Experiments

### 4.1 Experimental Setup

**Datasets.** We use Meta-Dataset (Triantafillou et al., 2019) – the most comprehensive and challenging large-scale FSL benchmark. It has 10 sub-datasets such as ImageNet (Deng et al., 2009) and Omniglot (Lake et al., 2015), with various domain gaps. Our experiments are conducted under the single training source setting, i.e. only ImageNet is used for training, and the meta-test split of all ten datasets for evaluation. Some of the test datasets such as CUB share similar or highly-related categories with ImageNet, while the others have greater domain gaps. Note that Hu et al. (2022) claims pretraining on all images in the training set of ImageNet is reasonable for introducing extra data and boosting the performance. However, such a strategy utilizes much more training samples (1.28M images, 1000 classes in ImageNet v.s. 0.9M images, 712 classes in meta-train split of ImageNet). Empirically so many additional images can greatly benefit generalization ability of self-supervised learning methods. Therefore to make a more fair comparison, we strictly follow the experiment protocol used in CTX (Doersch et al., 2020) and shall not use any extra data even in the unsupervised pretraining stage. We resize all images to $224 \times 224$ for ViT-small and $84 \times 84$ for ViT-tiny.

**Implementation details.** We set the patch size as 8 for ViT-tiny (as it has small input image size), and keep the other hyper-parameters as default. We adopt standard ViT-small with 12 layers, 6 attention heads, feature dimension as 384 and patch size as 16. We strictly follow the hyper-parameter setting and data augmentation in DINO (Caron et al., 2021) for pretraining. In test-time finetuning, we empirically set the hidden dimension $d'$ of the transformation module as $d/2$, and output dimension $d_{proj}$ of the projector as 64 for all datasets. We utilize AdamW optimizer finetuning, with learning rate set as $1e-3$ for TrafficSign and $5e-4$ for other datasets. $\lambda$ is set as 0.1. For simplicity, the selection of hyper-parameters is conducted on the meta-validation set of ImageNet, which is the only within-domain setting in Meta-Dataset.

**Evaluation benchmark.** We report the accuracy of randomly sampled 600 episodes for each dataset and the average accuracy when comparing with the existing methods. The comprehensive comparison of both accuracy and 95% confidence interval is in Appendix.

| Backbone | Image size | Params(M) | FLOPs(G) |
|---|---|---|---|
| Res18 | 84×84 | 11.69 | 1.82 |
| ViT-tiny | 84×84 | 5.38 | 0.72 |
| Res34 | 224×224 | 21.80 | 3.68 |
| ViT-small | 224×224 | 21.97 | 4.61 |

Table 1: Comparison of parameter size and FLOPs between different backbones.

### 4.2 Comparison with State-of-the-art Methods

Before the comprehensive comparison, it is necessary to show the comparison between different backbone is fair enough since our backbone model is not the same as the existing method. Therefore we present the comparison of size of model parameters and FLOPs in Tab. 1, in which the FLOPs of all models are computed by fvcore[1]. The results show that (1) compared with Res18, ViT-tiny is a much smaller and efficient model, and (2) ViT-small is approximately comparable to Res34. In this way, the comparison of our proposed method with state-of-the-art methods is reasonable and fair.

We compare our model with ProtoNet(Snell et al., 2017), CTX (Doersch et al., 2020), TSA (Li et al., 2022), etc. These methods take the backbones of ResNet18 or ResNet34. Also, the pretrain-meta-train-finetune baseline (P>M>F) (Hu et al., 2022) is not considered in computing average rank since extra data is used. As in Tab. 2, when using ViT-small as backbone whose parameter size is comparable to that of ResNet34, our model receives 1.6 average rank on all dataset. Specifically, on Texture and Fungi, our model outperforms the strongest competitors CTX and TSA by about 8% and 10%, while on other datasets the performance of our model is still comparable with or slight better than that of the existing methods. We notice that our model

---

[1]https://github.com/facebookresearch/fvcore

| Model | Backbone | ILSVRC | Omni | Acraft | CUB | DTD | QDraw | Fungi | Flower | Sign | COCO | Avg | Rank |
|---|---|---|---|---|---|---|---|---|---|---|---|---|---|
| Finetune | | 45.78 | 60.85 | 68.69 | 57.31 | 69.05 | 42.60 | 38.20 | 85.51 | 66.79 | 34.86 | 56.96 | 10.2 |
| Proto | | 50.50 | 59.98 | 53.10 | 68.79 | 66.56 | 48.96 | 39.71 | 85.27 | 47.12 | 41.00 | 56.10 | 10.5 |
| Relation | Res18 | 34.69 | 45.35 | 40.73 | 49.51 | 52.97 | 43.30 | 30.55 | 68.76 | 33.67 | 29.15 | 42.87 | 14.6 |
| P-MAML | | 49.53 | 63.37 | 55.95 | 68.66 | 66.49 | 51.52 | 39.96 | 87.15 | 48.83 | 43.74 | 57.52 | 9.2 |
| BOHB | | 51.92 | 67.57 | 54.12 | 70.69 | 68.34 | 50.33 | 41.38 | 87.34 | 51.80 | 48.03 | 59.15 | 8.2 |
| TSA | | **59.50** | **78.20** | 72.20 | **74.90** | 77.30 | 67.60 | 44.70 | 90.90 | 82.50 | **59.00** | **70.68** | 4.3 |
| Ours | ViT-t | 56.40 | 72.52 | **72.84** | 73.79 | **77.57** | 67.97 | **51.23** | 93.30 | 84.09 | 55.68 | 70.54 | 4.1 |
| Proto | | 53.70 | 68.50 | 58.00 | 74.10 | 68.80 | 53.30 | 40.70 | 87.00 | 58.10 | 41.70 | 60.39 | 7.4 |
| CTX | Res34 | 62.76 | 82.21 | 79.49 | 80.63 | 75.57 | **72.68** | 51.58 | 95.34 | 82.65 | 59.90 | 74.28 | 2.8 |
| TSA | | 63.73 | **82.58** | **80.13** | 83.39 | 79.61 | 71.03 | 51.38 | 94.05 | 81.71 | 61.67 | 74.93 | 2.5 |
| P>M>F* | ViT-s | 74.69 | 80.68 | 76.78 | 85.04 | 86.63 | 71.25 | 54.78 | 94.57 | 88.33 | 62.57 | 77.53 | — |
| Ours | | **67.37** | 78.11 | 79.94 | **85.93** | **87.62** | 71.34 | **61.80** | **96.57** | 85.09 | 62.33 | **77.61** | **1.6** |

Table 2: Test accuracies and average rank on Meta-Dataset. Note that different backbones are adopted by these methods. * denotes using extra data for training. The bolded items are the best ones with highest accuracies.

is inferior to the best ones in Omniglot, while this is reasonable. Since Omniglot images represent simple characters with monotone color patterns, each image patches contain less information than images in other datasets. Vanilla ViTs have less efficiency in dealing with these image patches due to limited interaction among patch embeddings. This problem can be solved with much sophisticated variants of ViT like Swin (Liu et al., 2021b), and will be taken as future works. Moreover, our proposed method is better than P>M>F, which not only utilizes extra data for training but also finetunes all model parameters during testing, on more than half of the datasets, which strongly indicates the effectiveness of the proposed finetuning strategy in this paper. As for using ViT-tiny which has much less parameter than Res18, our model is still comparable to the state-of-the-art methods and outperforms many popular baselines. Particularly, compared with ProtoNet which is one of the most famous and efficient methods in FSL, our eTT shows significant boost on Aircraft by 19.74% and TrafficSign by 36.97%. The reason of the inferior results on several datasets against TSA can be two folds. Firstly, the ViT-tiny intrinsically has smaller capacity than Res18. On the other hand, while it is common to train ViT with large scale images and patches so that the images are splitted into abundant patches and each patch-level token can receive enough information. In contrast, we adopt $84 \times 84$ images with $8 \times 8$ patch size for ViT-tiny so that the comparison with Res18 is fair, which lead to less patches with smaller size and may have negative influence on the performance. In general, the results indicate that our proposed eTT can make ViT models a desirable choice for large scale FSL problems.

## 4.3 Model Analysis

To further validate the effectiveness of our method, we conduct a series of ablation studies on Meta-Dataset using ViT-small below.

### 4.3.1 Design of Each Module

**Can finetuning on meta-train set boost the performance?** One would ask whether it is necessary to make use of base annotations, as supervised pretraining is also effective in many FSL works (Ye et al., 2020; Hou et al., 2019). To verify it, we finetune DINO-pre-trained ViT-small on meta-train split of ImageNet, in which the options of all hyper-parameters and data augmentations follow DeiT (Touvron et al., 2021) using either way of class token features or averaged patch features as image representations. After such a supervised finetuning, we test the models both with the basic test-time finetuning method as in Sec. 3.3, which we denote as LT+NCC, and with our proposed eTT. The results are shown in Fig. 3, from which we find out that (1) Supervised finetuning does improve test accuracies on ImageNet, CUB and MSCOCO. Particularly, the token finetune model receives 89.83% accuracy on CUB when testing with our eTT, which is remarkably better than any other models. This is reasonable as similar images between ImageNet and these datasets. By

| Model | ILSVRC | Omni | Acraft | CUB | DTD | QDraw | Fungi | Flower | Sign | COCO | Avg |
|---|---|---|---|---|---|---|---|---|---|---|---|
| Proto | 63.37 | 65.86 | 45.11 | 72.01 | 83.50 | 60.88 | 51.02 | 92.39 | 49.23 | 54.99 | 63.84 |
| LT+NCC | 65.96 | 67.62 | 64.03 | 77.10 | 83.46 | 63.88 | 57.79 | 93.13 | 66.91 | 56.04 | 69.59 |
| Last | 66.32 | 71.04 | 78.04 | 86.25 | 86.67 | 64.22 | 55.69 | 94.44 | 65.55 | 55.94 | 72.42 |
| First | 61.54 | 50.46 | 69.23 | 79.17 | 83.10 | 68.69 | 49.93 | 93.50 | 54.28 | 58.45 | 66.84 |
| LN | 66.22 | 70.45 | 69.41 | 81.29 | 86.37 | 66.28 | 58.38 | 96.25 | 71.09 | 59.57 | 72.53 |
| APT | 66.75 | 75.16 | 75.41 | 84.25 | 86.47 | 69.55 | 60.03 | 96.38 | 78.20 | 61.10 | 75.33 |
| Adapter | 66.53 | 72.31 | 73.75 | 83.73 | 86.86 | 66.74 | 58.49 | 96.15 | 82.65 | **62.40** | 74.93 |
| eTT | **67.37** | **78.11** | **79.94** | **85.93** | **87.62** | **71.34** | **61.80** | **96.57** | **85.09** | 62.33 | **77.61** |
| Random | 66.12 | 76.33 | 78.35 | 84.77 | 86.78 | 70.13 | 59.25 | 96.00 | 82.28 | 59.59 | 75.96 |
| Avg | 66.11 | 75.06 | 77.07 | 85.16 | 87.35 | 70.72 | 61.79 | 96.54 | 84.28 | 62.18 | 76.73 |
| Sampling | 67.81 | 76.72 | 77.96 | 85.79 | 87.25 | 70.19 | 60.73 | 96.27 | 83.72 | 62.17 | 76.86 |
| Full | **67.37** | **78.11** | **79.94** | **85.93** | **87.62** | **71.34** | **61.80** | **96.57** | **85.09** | **62.33** | **77.61** |
| Linear | 66.35 | 74.26 | 79.42 | 83.65 | 86.02 | 71.11 | 55.73 | 95.89 | 82.73 | 59.90 | 75.51 |
| Bottleneck | 67.29 | 76.06 | 79.72 | 85.60 | 87.21 | 70.59 | 61.59 | 96.15 | 85.00 | 62.02 | 77.12 |
| FiLM | 66.91 | 75.32 | 78.26 | 85.78 | 86.83 | 70.29 | 61.65 | 96.50 | 84.48 | 61.75 | 76.78 |
| Offset | **67.37** | **78.11** | **79.94** | **85.93** | **87.62** | **71.34** | **61.80** | **96.57** | **85.09** | **62.33** | **77.61** |
| w/o PR | 66.72 | 74.20 | 78.42 | 85.06 | 87.01 | 70.34 | 61.64 | 96.51 | 84.23 | 61.08 | 76.52 |
| w PR | **67.37** | **78.11** | **79.94** | **85.93** | **87.62** | **71.34** | **61.80** | **96.57** | **85.09** | **62.33** | **77.61** |
| w/o Stand | 67.09 | 76.42 | 78.87 | 83.10 | 86.50 | 70.09 | 61.02 | 96.33 | 82.88 | 61.33 | 76.36 |
| w Stand | **67.37** | **78.11** | **79.94** | **85.93** | **87.62** | **71.34** | **61.80** | **96.57** | **85.09** | **62.33** | **77.61** |

Table 3: Test accuracies on Meta-Dataset of different variants of our proposed method. The bolded items are the best ones with highest accuracies.

training on the image annotations of ImageNet, the model learns class-specific knowledge which cannot be obtained during self-supervised learning. Since the categories are highly correlated and overlapped among these datasets, the learned knowledge can also benefit the recognition on these novel datasets even though the specific novel classes do not appear in the meta-train set. (2) Despite the improvement on the three datasets, models with supervised finetuning degrade on the other datasets, especially on Traffic Sign and VGG Flower. This is due to fitting class labels weakens the effect of these features and makes it harder to generalize to novel domains. When taking into account the performance of all datasets, pretraining with DINO is generally the much more desirable choice for better generalization over different domains. (3) The improvement of our propose method against the basic LT+NCC is not consistent among three different kinds of pretraining strategy. For example, while our method can boost the performance of DINO pre-trained model by 9.47% on Aircraft and 4.83% on CUB, it can only bring much less advantage on models with supervised finetuning.

**Effectiveness of APT and DRA.** We test the DINO pre-trained model with different kinds of testing strategies including (1) Proto: Directly generating predictions based on ProtoNet. The prototypes are computed using averaged class token features from each category. (2) LT+NCC: The basic test-time finetuning method in Sec. 3.3. (3) Last: Finetuning the last transformer layer during testing, together with LT+NCC. which has similar parameter size to our method. (4) First: Finetuning the first transformer layer during testing, together with LT+NCC. which has similar parameter size to our method. (5) LN: We try to finetune the affinity parameter in each layer normalization as an alternative finetune strategy, which is utilized in many cross-domain FSL works (Tseng et al.; Tsutsui et al., 2022). (6) APT: The model is finetuned using APT together with LT+NCC, using cross entropy loss and the proposed prototypical regularization. (7) Adapter: The model is finetuned using DRA together with LT+NCC, using cross entropy loss. (8) eTT: The model is finetuned using our proposed APT, DRA and LT+NCC. The results in Tab. 3 show that while LT+NCC can fundamentally improve the model which indicates the importance of test-time finetuning, adding our proposed modules to the finetuning procedure can consistently bring higher performance. Also, finetuning specific transformer layer can only bring limited improvement on few datasets: finetuning the last

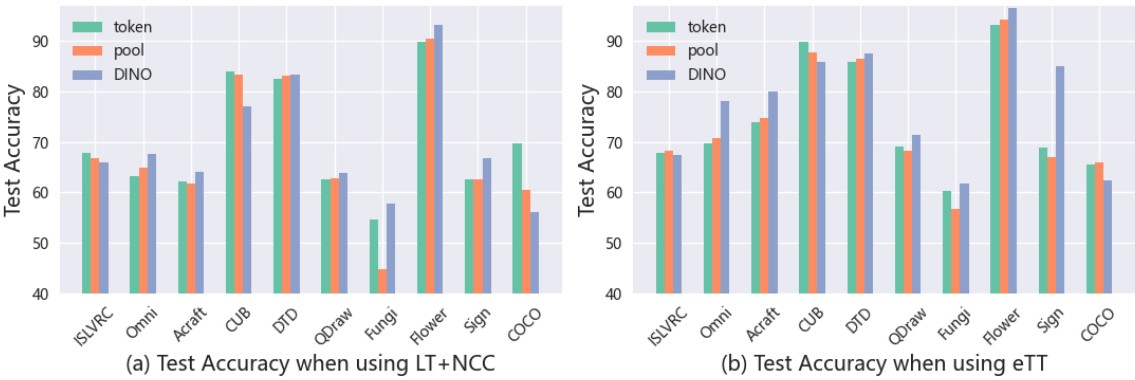

Figure 3: Test accuracy of different training strategy if testing with (a) LT+NCC or (b) our eTT.

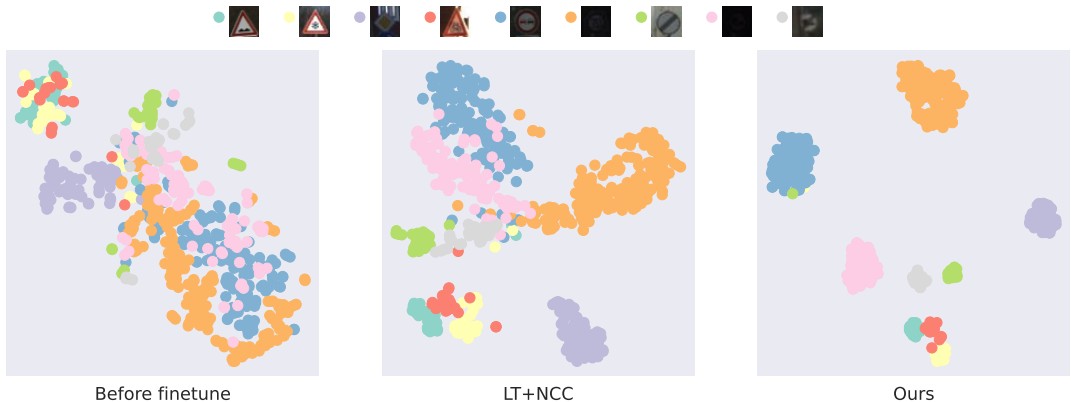

Figure 4: Visualization of feature embeddings from a randomly sampled episode of TrafficSign.

layer leads to good performance on Aircraft, CUB and Texture, while updating the first layer leads to good performance on Quickdraw and MSCOCO. However, this simple finetuning strategy cannot bring consistent improvement on all datasets. This indicates that different data requires different levels of adaptation, and the improvement is much smaller than that of our method. Moreover, we give the tSNE visualization of feature embeddings of a randomly sampled episode from TrafficSign in Fig. 4, which demonstrates that utilizing our proposed method can better regulate the feature embeddings into proper clusters.

**Is prototypical initialization necessary?** One of the most important parts of our APT is the attentive prototypical initialization in which we use attentively aggregated patch embeddings to initialize the prefix matrix. To verify this strategy, we compare several different choices of initialization, including (1) Random: random initialization from normal distribution. (2) Avg: simply averaging all patch embeddings from each category. (3) Sampling: randomly sampling one image for each category, and then initializing the prefix matrix with the averaged patch embeddings of each image. (4) Full: computing prototypes with our proposed attentive prototype. Results in Tab. 3 show that random initialization performs the worst, which can be resulted from insufficient task-specific information provided by the prefix in this way. Meanwhile, among all other strategies, using the attention map to aggregate patch embeddings as in Eq. 5 is better than simply averaging, leading to about 1% improvement on average.

**Do we need a more complex adapter structure?** One would argue that our DRA structure is too simple to learn the complex knowledge from support images. In Tab. 3 we compare three different instantiations of adapters including (1) Linear: As in Li et al. (2022), we use a linear layer for each adapter, whose output are than added to the original features in the MSA and FFN. (2) Bottleneck: We expand the linear layer

to a bottleneck structure where two linear layers are used for each adapters. (3) FiLM: We compare DRA with a FiLM-like variant, in which we add a scaling vector for each adapter as in FiLM layer Perez et al. (2018). Note that such a method is similar to MTL (Sun et al., 2019) in FSL. The difference lies in that we still use the original way to directly tune the parameters on the novel support sets, instead of using another meta-trained module to generate the parameters. (4) Offset: Only an offset vector is adopted for each adapter. The results reveals that the linear adapter performs the worst, which means such a structure is improper for ViT in finetuning. Moreover, we also find that using the bottleneck adapter will result in a dilemma. If we use small initial value for the adapter, the weights of each layer can only achieve gradient with extremely small values. As the result, these weights, except the bias term of the last layer, can hardly be updated based on the support knowledge, which means such an architecture almost equals to our design where only an offset vector is utilized. On the other hand, if large initial values are adopted to avoid gradient diminishing, then the output features from the adapters can make the predictions severely deviate from those without adapters, thus leading to worse performance. As for the FiLM-like DRA, it is worse than offset DRA by about 0.8% on average, while it doubles the parameter size based on offset DRA, leading to no significant additional efficacy.

**Effectiveness of prototypical regularization.** We also validate this regularization. In Tab. 3 we present the test accuracy when finetuning with and without this loss function. We can find that by applying this objective function, the model can have higher results on most datasets. Besides, as described in Sec 3.6, we use a standardization technique when computing the prototypical regularization. To verify its efficacy, we compare the model with and without such a standardization. The results are shown in Tab. 3. When not using standardization, the results are generally worse given comparable confidence intervals (Tab. 11). The results verify that this strategy can help the model with more stable finetuning procedure.

| $d_{proj}$ | ILSVRC | Omni | Acraft | CUB | DTD | QDraw | Fungi | Flower | Sign | COCO | Avg |
|---|---|---|---|---|---|---|---|---|---|---|---|
| 64 | 67.18 | 75.30 | 78.88 | 86.20 | 87.09 | 69.82 | 61.61 | 96.31 | 82.24 | 62.14 | 76.68 |
| 96 | 66.23 | 75.69 | 78.26 | 85.67 | 87.28 | 70.25 | **61.97** | **96.59** | 84.10 | 62.17 | 76.82 |
| 128 | 67.31 | 76.83 | 78.81 | 85.77 | 87.36 | 70.16 | 60.81 | 96.53 | 84.29 | 62.12 | 77.00 |
| 256 | 66.83 | 78.04 | 78.38 | 84.60 | 86.68 | 70.43 | 61.03 | 96.23 | **85.33** | 62.10 | 76.97 |
| 192 | **67.37** | **78.11** | **79.94** | **85.93** | **87.62** | **71.34** | 61.80 | 96.57 | 85.09 | **62.33** | **77.61** |

Table 4: Test accuracies on Meta-Dataset of different variants of our proposed method. The bolded items are the best ones with highest accuracies.

### 4.3.2 Comparison among Different Hyper-parameter Settings

In additional to the ablation study about the proposed module, We further verify different choices of hyper-parameters in our model. Especially, $d_{proj}$ for the transformation module in APT and $\lambda$ for the prototypical regularization are tested in Tab. 4 and Tab. 5 in the Appendix. In general, the improvement is not consistent. For $d_{proj}$, we can find that using 192-d hidden dimension can get globally better results, which indicates that such a choice can make a good balance between the model capacity and scale so that the finetuning can be conducted both efficiently and effectively. As for $\lambda$, 0.1 seems to be a desirable choice. Intuitively, smaller $\lambda$ leads to less control of the prefix from the proposed prototypical regularization. Therefore, the prefix may lose the desired information during the optimization on the support set. On the other hand, when $\lambda$ is too large, the regularization overwhelms the label supervision, and thus the model can hardly adapt to the support knowledge, leading to worse performance especially on Omniglot and Aircraft.

## 5 Conclusion

We propose a novel finetuning method named efficient Transformer Tuning (eTT) for few-shot learning with ViT as our backbone. By fixing the parameters in the backbone and utilizing attentive prefix tuning and domain residual adapter, our method can guide the ViT model with comprehensive task-specific information, which leads to better representations and performance. This is demonstrated by the fact that we establish new state-of-the-arts on the large-scale benchmark Meta-Dataset.

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

# A   Appendix

## A.1   Limitations and Future Work

Despite the marginal effectiveness and efficiency of our proposed eTT, we mainly notice two points that should be explored in the future: (1) The plain ViT backbone utilized in this paper, may not be the best choice to the simple dataset, e.g., Omniglot, while a well-designed ViT backbone may potentially better improve the efficacy of our method on such dataset. (2) A flexible finetuning algorithm such as (Lee et al.) may have better generalization ability when facing episodes with various shots and ways, than the commonly-used methods that adopt fixed test-time finetuning iterations. However, it is non-trivial to directly merge such methods with our proposed eTT due to different network structures and tuning strategies. It can be taken as the future work to properly utilize these flexible finetuning algorithm to further improve the performance of ViT in FSL.

### A.2 Additional Experiment Results

#### A.2.1 Full Comparison with state-of-the-art methods

We show the accuracies together with confidence interval in Tab. 6. Beyond the accuracies which is analyzed in the main context, the confidence interval of our eTT on both ViT-tiny and ViT-small is comparable with the other competitors, which reflects that our method is stable and robust enough among different testing episodes.

#### A.2.2 Influence of Training set

As we have stated in the main context, our eTT is trained on the meta-train split of ImageNet to make fair comparison with other methods. To show to what extent training on the full ImageNet training set instead of the meta-train set can impact the performance, we test our eTT using off-the-shelf DINO ViT-s model. The results are shown in Tab. 7. We can find that (1) For those datasets on which DINO meta-train performs better than P>M>F, using full ImageNet to train DINO can bring further improvement. (2) With the help of more data, our eTT overpasses P>M>F on ILSVRC and MSCOCO. (3) While more data does improve the results on Omniglot and TrafficSign, the final results are still worse than those of P>M>F, which we think may be correlated with the limitations of our method as analyzed above. Given all these results, as a lighter model in that no meta-training phase is utilized and only few parameters are engaged in the test-time tuning, our method can still enjoy comparable performance with P>M>F when training on same amount of data.

#### A.2.3 Influence of DINO

The DINO pretrain procedure is an important part of our method. To verify the effectiveness of DINO pretrain so that the comparison with other methods is fair enough, we present in Tab. 8 the results of TSA using DINO-pretrained ResNet-34 and eTT using supervised pretrained ViT-s. We can find that (1) The effect of DINO is not consistent on two backbones. While DINO benifits our eTT with about 5% accuracy on average, it severely weakens the performance of TSA with a large margin. It means that for FSL, different backbones require different pretrain strategy respectively. (2) While supervised pretrained ViT-s performs worse on most datasets, it is better on CUB and COCO, which indicates learning label information from ImageNet can help the model understanding novel knowledge from these two datasets.

#### A.2.4 Verification of potential overfitting in finetuning

As we have stated in Sec. 1, finetuning the whole backbone model with few support data will meet potential overfitting problem. To reveal if such a problem exists in Meta-Dataset, we conduct an experiment as follow: during a normal testing phase, we select all episodes whose minimum shot (minimum number of support images for each class) is no larger than 2 (extremely small number of labelled instances), and compare the average accuracies of eTT and simple finetuning based on these episodes. Tab. 9 and Tab. 10 show the accuracies on support sets and query sets respectively. We can find that most of the datasets both methods receive nearly 100% accuracy, which means these two methods can well learn the support data. Given this fact, finetuning is much worse than eTT in terms of query accuracies. The overfitting can be reflected given high training accuracies and worse testing performance, and to some extent our proposed eTT can fix this problem.

#### A.2.5 More Visualization

We visualize the self-attention map from models with and without DRA on ILSVRC and TraffignSign in Fig. 5. Specifically, we randomly sample an episode from each dataset and use our eTT to tune the model based on the support samples. Then we calculate the self-attention map of the last layer's class token and highlight the areas with top 20% attention scores. We can find that for the in-domain ILSVRC episode, the model can attend to similar regions no matter whether DRA is used. In contrast, the model without DRA can easily attend to background regions with less valuable information, which reveals a potential reason that these two models has similar accuracies on ILSVRC but large performance gap on TrafficSign.

Futhermore, we visualize more episodes from Aircraft, TrafficSign and MSCOCO in Fig. 6, Fig. 7 and Fig. 8, which shows that our porposed eTT can remarkably improve the embedding space after test-time finetuning.

## A.3 Broader Impact

Our paper presents a more efficient and practical FSL pipeline utilizing ViT. We hope this work can shed light on the broader usage of ViT in FSL tasks. On the other hand, the proposed method can provide researchers with alternative choice for FSL applications in real-case scenarios with large-scale meta-train set and challenging various test episodes.

| $d_{proj}$ | ILSVRC | Omni | Acraft | CUB | DTD | QDraw | Fungi | Flower | Sign | COCO | Avg |
|---|---|---|---|---|---|---|---|---|---|---|---|
| 0.01 | 67.01 | 76.56 | 78.34 | 85.53 | 86.96 | 70.03 | 61.20 | 96.17 | 85.00 | 62.67 | 76.95 |
| 0.05 | 66.49 | 77.40 | 78.92 | 85.80 | 87.54 | 70.23 | 60.78 | 96.28 | 84.95 | 62.38 | 77.08 |
| 0.5 | 66.88 | 77.73 | 78.65 | **86.00** | 87.15 | 70.48 | 61.64 | 96.23 | 84.39 | 63.39 | 77.25 |
| 0.9 | 67.03 | 76.55 | 77.89 | 85.78 | 87.04 | 70.08 | **62.45** | 96.20 | 84.44 | **62.83** | 77.03 |
| 0.1 | **67.37** | **78.11** | **79.94** | 85.93 | **87.62** | **71.34** | 61.80 | **96.57** | **85.09** | 62.33 | **77.61** |

Table 5: Test accuracies on Meta-Dataset of different variants of our proposed method. The bolded items are the best ones with highest accuracies.

| Model | Backbone | ILSVRC | Omni | Acraft | CUB | DTD | QDraw | Fungi | Flower | Sign | COCO | Rank |
|---|---|---|---|---|---|---|---|---|---|---|---|---|
| Finetune | | $45.78_{1.10}$ | $60.85_{1.58}$ | $68.69_{1.26}$ | $57.31_{1.26}$ | $69.05_{0.90}$ | $42.60_{1.17}$ | $38.20_{1.02}$ | $85.51_{0.68}$ | $66.79_{1.31}$ | $34.86_{0.97}$ | 10.2 |
| Proto | | $50.50_{1.08}$ | $59.98_{1.35}$ | $53.10_{1.00}$ | $68.79_{1.01}$ | $66.56_{0.83}$ | $48.96_{1.08}$ | $39.71_{1.11}$ | $85.27_{0.77}$ | $47.12_{1.10}$ | $41.00_{1.10}$ | 10.5 |
| Relation | Res18 | $34.69_{1.01}$ | $45.35_{1.36}$ | $40.73_{0.83}$ | $49.51_{1.05}$ | $52.97_{0.69}$ | $43.30_{1.08}$ | $30.55_{1.04}$ | $68.76_{0.83}$ | $33.67_{1.05}$ | $29.15_{1.01}$ | 14.6 |
| P-MAML | | $49.53_{1.05}$ | $63.37_{1.33}$ | $55.95_{0.99}$ | $68.66_{0.96}$ | $66.49_{0.83}$ | $51.52_{1.00}$ | $39.96_{1.14}$ | $87.15_{0.69}$ | $48.83_{1.09}$ | $43.74_{1.12}$ | 9.2 |
| BOHB | | $51.92_{1.05}$ | $67.57_{1.21}$ | $54.12_{0.90}$ | $70.69_{0.90}$ | $68.34_{0.76}$ | $50.33_{1.04}$ | $41.38_{1.12}$ | $87.34_{0.59}$ | $51.80_{1.04}$ | $48.03_{0.99}$ | 8.2 |
| TSA | | $59.50_{1.10}$ | $78.20_{1.20}$ | $72.20_{1.00}$ | $74.90_{0.90}$ | $77.30_{0.70}$ | $67.60_{0.90}$ | $44.70_{1.00}$ | $90.90_{0.60}$ | $82.50_{0.80}$ | $59.00_{1.00}$ | 4.3 |
| Ours | ViT-t | $56.40_{1.13}$ | $72.52_{1.36}$ | $72.84_{1.04}$ | $73.79_{1.09}$ | $77.57_{0.84}$ | $67.97_{0.88}$ | $51.23_{1.15}$ | $93.30_{0.57}$ | $84.09_{1.07}$ | $55.68_{1.05}$ | 4.1 |
| Proto | | $53.70_{1.07}$ | $68.50_{1.27}$ | $58.00_{0.96}$ | $74.10_{0.92}$ | $68.80_{0.77}$ | $53.30_{1.06}$ | $40.70_{1.15}$ | $87.00_{0.73}$ | $58.10_{1.05}$ | $41.70_{1.08}$ | 7.4 |
| CTX | Res34 | $62.76_{0.99}$ | $82.21_{1.00}$ | $79.49_{0.89}$ | $80.63_{0.88}$ | $75.57_{0.64}$ | $72.68_{0.82}$ | $51.58_{1.11}$ | $95.34_{0.37}$ | $82.65_{0.76}$ | $59.90_{1.02}$ | 2.8 |
| TSA | | $63.73_{0.99}$ | $82.58_{1.11}$ | $80.13_{1.01}$ | $83.39_{0.80}$ | $79.61_{0.68}$ | $71.03_{0.84}$ | $51.38_{1.17}$ | $94.05_{0.45}$ | $81.71_{0.95}$ | $61.67_{0.95}$ | 2.5 |
| Ours | ViT-s | $67.37_{0.97}$ | $78.11_{1.22}$ | $79.94_{1.06}$ | $85.93_{0.91}$ | $87.62_{0.57}$ | $71.34_{0.87}$ | $61.80_{1.06}$ | $96.57_{0.46}$ | $85.09_{0.90}$ | $62.33_{0.99}$ | 1.6 |

Table 6: Test accuracies, confidence interval and average rank on Meta-Dataset. Note that different backbones are adopted by these methods. The bolded items are the best ones with highest accuracies.

| Model | Train Set | ILSVRC | Omni | Acraft | CUB | DTD | QDraw | Fungi | Flower | Sign | COCO | Avg |
|---|---|---|---|---|---|---|---|---|---|---|---|---|
| eTT | meta-train | 67.37 | 78.11 | 79.94 | 85.93 | 87.62 | **71.34** | 61.80 | 96.57 | 85.09 | 62.33 | 77.61 |
| eTT | full | **74.76** | 78.73 | **80.10** | **86.99** | **87.72** | 71.20 | **61.95** | **96.66** | 85.83 | **64.25** | **78.82** |
| P>M>F | full | 74.69 | **80.68** | 76.78 | 85.04 | 86.63 | 71.25 | 54.78 | 94.57 | **88.33** | 62.57 | 77.53 |

Table 7: Test accuracies on Meta-Dataset of different variants of our proposed method. The bolded items are the best ones with highest accuracies. The highlighted rows denote the final model in our main paper.

| Model | Pretrain | ILSVRC | Omni | Acraft | CUB | DTD | QDraw | Fungi | Flower | Sign | COCO | Avg |
|-------|----------|--------|------|--------|-----|-----|-------|-------|--------|------|------|-----|
| TSA | Sup. | 59.50 | **78.20** | 72.20 | 74.90 | 77.30 | 67.60 | 44.70 | 90.90 | 82.50 | 59.00 | 70.68 |
| TSA | DINO | 48.18 | 64.94 | 56.74 | 45.49 | 69.06 | 59.51 | 31.13 | 81.01 | 48.70 | 26.18 | 53.09 |
| eTT | Sup. | 65.17 | 67.47 | 73.30 | 87.71 | 84.50 | 67.46 | 55.51 | 92.55 | 64.08 | 63.68 | 72.14 |
| eTT | DINO | **67.37** | 78.11 | **79.94** | **85.93** | **87.62** | **71.34** | **61.80** | **96.57** | **85.09** | **62.33** | **77.61** |

Table 8: Test accuracies on Meta-Dataset of different variants of our proposed method. The bolded items are the best ones with highest accuracies. The highlighted rows denote the final model in our main paper.

| Model | ILSVRC | Omni | Acraft | CUB | DTD | QDraw | Fungi | Flower | Sign | COCO | Avg |
|-------|--------|------|--------|-----|-----|-------|-------|--------|------|------|-----|
| eTT | 100.00 | 99.99 | 100.00 | 100.00 | 100.00 | 100.00 | 99.79 | 100.00 | 100.00 | 99.15 | 99.89 |
| FT | 100.00 | 99.87 | 100.00 | 100.00 | 100.00 | 100.00 | 96.95 | 100.00 | 100.00 | 95.20 | 99.20 |

Table 9: Support set accuracies of eTT and Finetune on testing episodes whose minimum shots is no larger than 2.

| Model | ILSVRC | Omni | Acraft | CUB | DTD | QDraw | Fungi | Flower | Sign | COCO | Avg |
|-------|--------|------|--------|-----|-----|-------|-------|--------|------|------|-----|
| FT | 29.19 | 54.54 | 35.10 | 41.54 | 53.66 | 43.37 | 38.53 | 76.76 | 72.90 | 41.21 | 48.68 |
| eTT | **40.22** | **64.79** | **41.33** | **55.11** | **66.20** | **49.14** | **56.33** | **85.03** | **75.29** | **56.19** | **58.96** |

Table 10: Query set accuracies of eTT and Finetune on testing episodes whose minimum shots is no larger than 2. The bolded items are the best ones with highest accuracies.

| Model | ILSVRC | Omni | Acraft | CUB | DTD | QDraw | Fungi | Flower | Sign | COCO |
|-------|--------|------|--------|-----|-----|-------|-------|--------|------|------|
| w/o stand | 1.06 | 1.25 | 1.05 | 0.89 | 0.64 | 0.92 | 1.05 | 0.38 | 0.96 | 0.96 |
| w stand | 0.97 | 1.22 | 1.06 | 0.91 | 0.57 | 0.87 | 1.06 | 0.46 | 0.90 | 0.99 |

Table 11: The corresponding confidence intervals of models in ablation study on standardization.

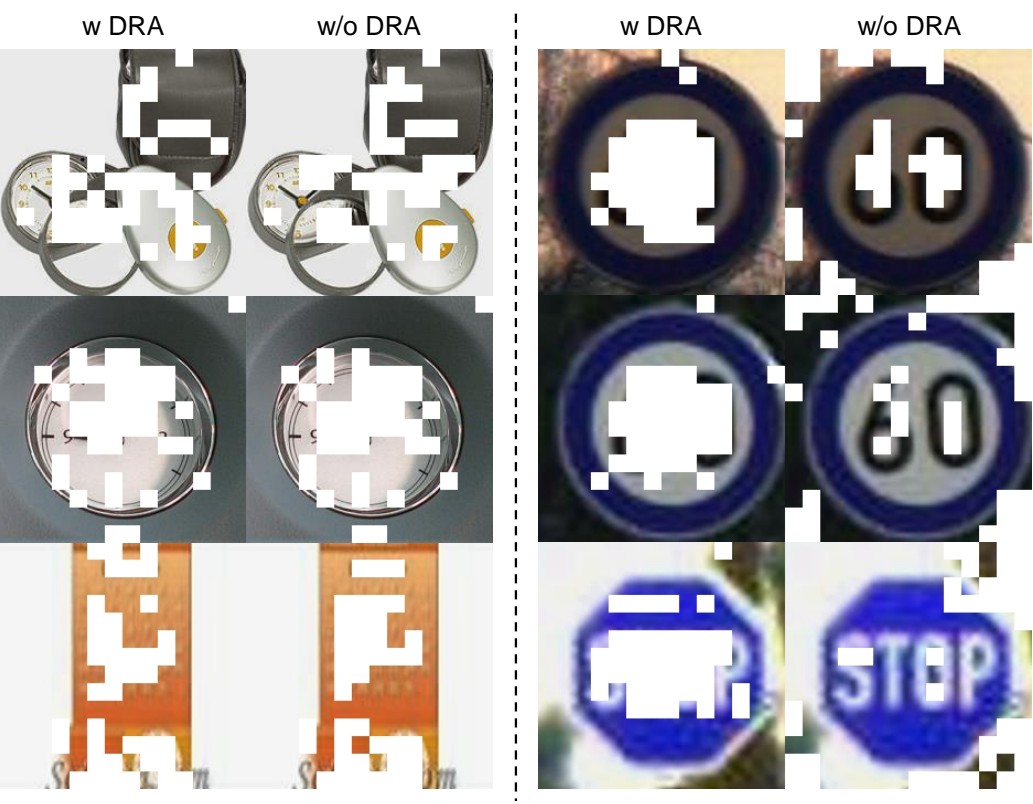

Figure 5: Visualization of self-attention from model with and without DRA on ILSVRC (left) and TrafficSign (right). The white regions are those with top 20% attention scores

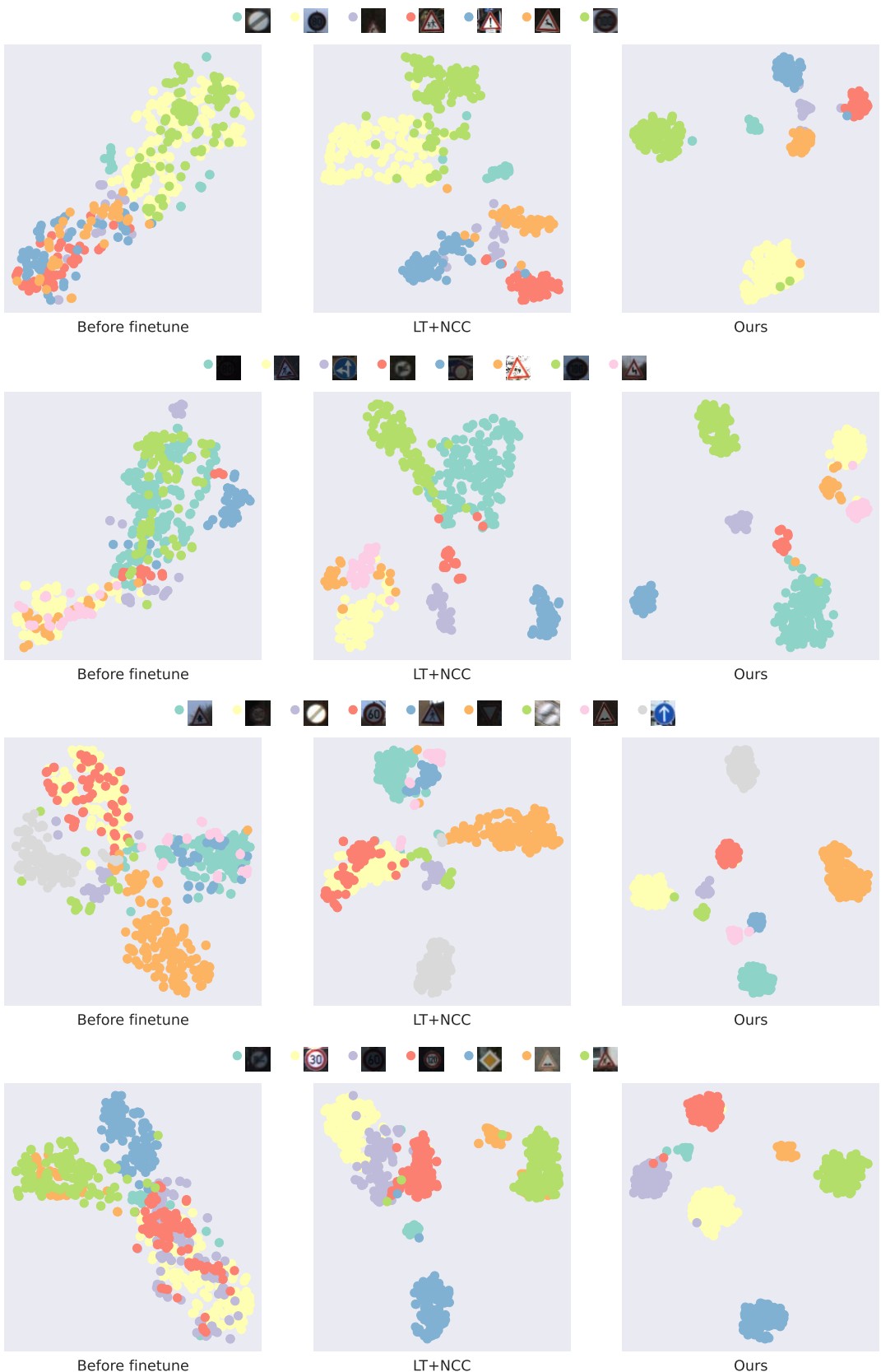

Figure 6: More visualization of feature embeddings from a randomly sampled episode of TrafficSign.

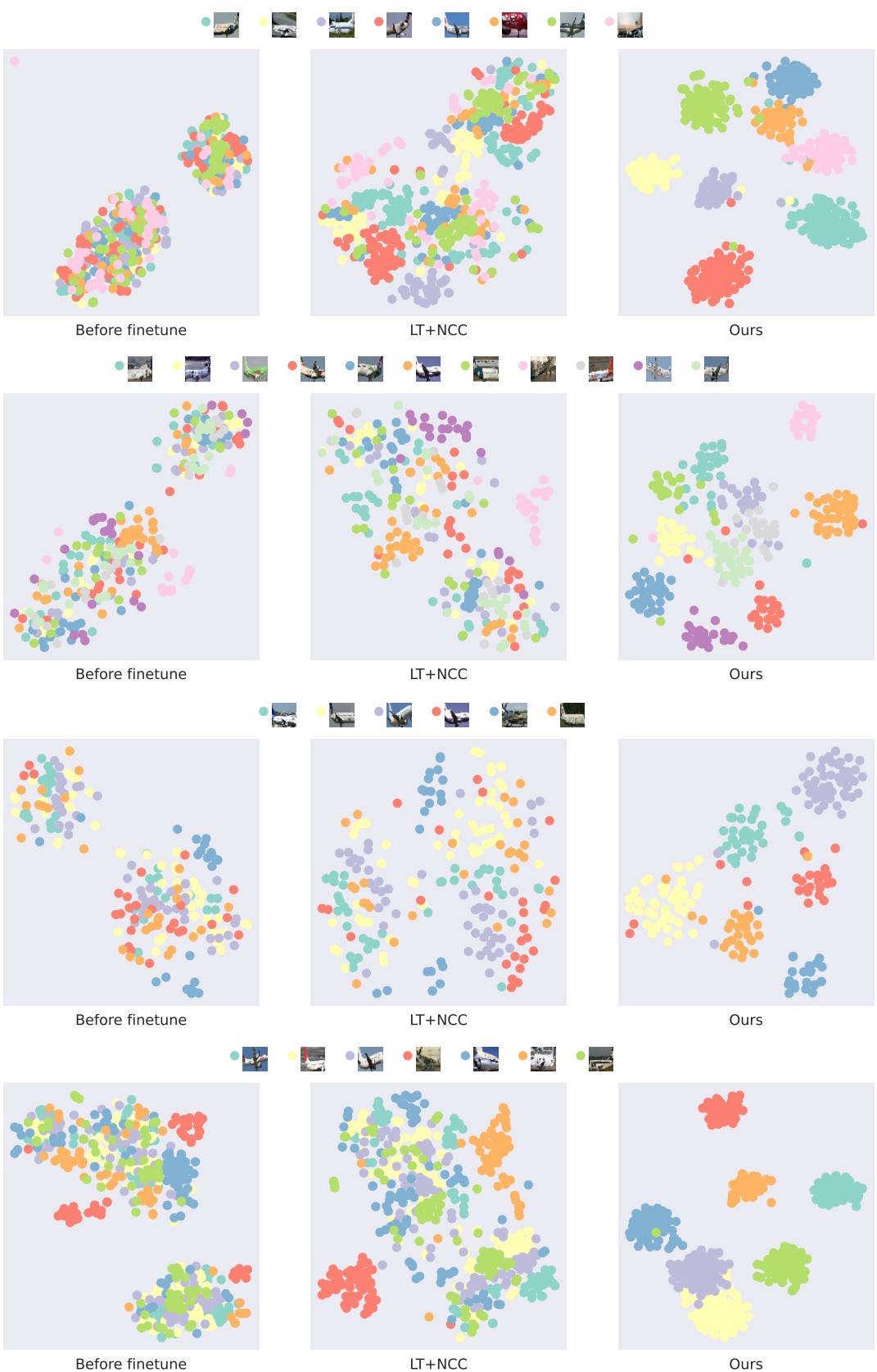

Figure 7: More visualization of feature embeddings from a randomly sampled episode of Aircraft.

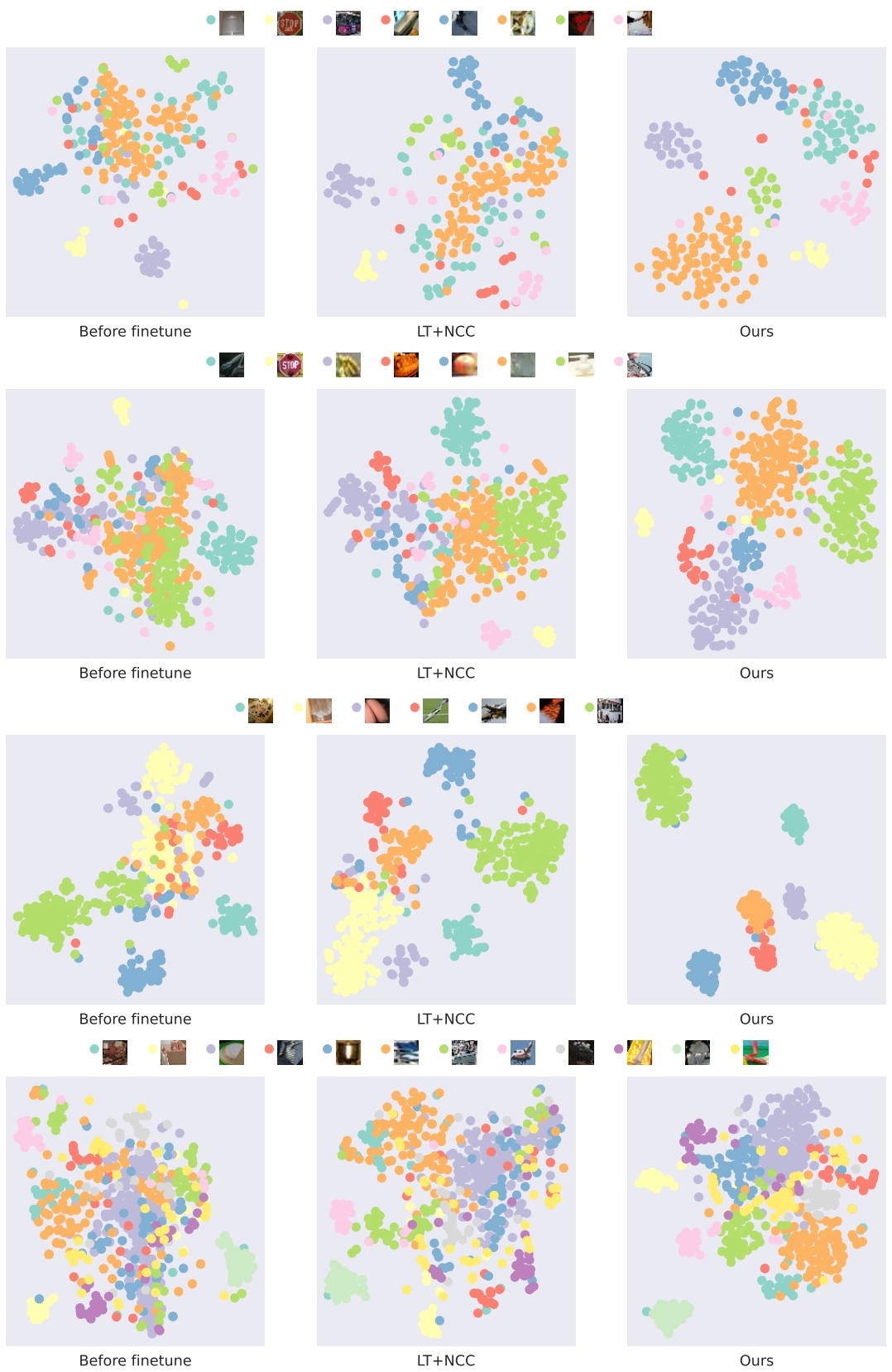

Figure 8: More visualization of feature embeddings from a randomly sampled episode of MSCOCO.

