# OpenReview forum: "Exploring Efficient Few-shot Adaptation for Vision Transformers"
_TMLR — Accepted by TMLR_

### Review · Reviewer_hyDZ · 2022-07-11

**Summary Of Contributions:**

This paper proposes an approach for adapting a pretrained ViT model for few-shot learning tasks. In contrast to prior work, their finetuning approach has a small number of trainable parameters.

In more detail, they introduce a learnable “prefix” (drawing inspiration from prefix tuning in NLP). In their case, this “prefix” is a matrix containing N_p d-dimensional vectors, where d is the dimensionality of the transformer. Then, a learnable layer maps those to a higher dimensionality, in order to yield N_p d-dimensional vectors for each key and each value matrix of each layer which are concatenated with the usual keys and values of that layer. This structure allows the task-specific information (captured in the prefix, which is learned for each new task) to affect all layers of the network. In practice N_p is set to N which is the number of classes in the given task, and the d-dimensional vectors are initialized in a specific way, from what they refer to as “attentive prototypes”. Specifically, the attention mask is computed between the class token (which contains global class information) and the final-layer patch embeddings from the transformer, for each support set image. This attention mask is then applied over image-level representations. A prototype per class is computed from these masked image-level representations by averaging over representations of (masked patches of images of) each class. The concatenation of these prototypes is the initialization to the prefix matrix. During training on each task, this prefix is learned, but they include a regularizer that incentivizes not to move too much from that initialization. Finally, they introduce a module called Domain Residual Adapter (DRA) which is comprised of learnable offset parameters for the MSA and FFN sub-layers of each transformer layer. This allows more flexibility to change the backbone, which they hypothesize is useful for tasks that are substantially different from the pretraining data.

Empirically, they use a pretrained DINO model with ViT-tiny and -small, and show improved results over baselines on Meta-Dataset (setting of training only on ImageNet). They also run a number of ablations and analyses to verify the usefulness of the different components of their method.


**Broader Impact Concerns:**

I don’t have any specific concerns on the ethical implications of this work.


**Requested Changes:**

# Critical changes

## Add missing related work

The authors erroneously claim that ViTs haven’t been used for few-shot learning, citing a single exception. A couple other papers that have used them are [1] and [2] (see References at the bottom).

The “Remark” paragraph in section 3.5 would be more appropriate to move into the related work section. Also, there are some related works missing from that context too. CNAPs [4], and FLUTE [5] also use a form a adapters (FiLM conditioning) and the latter is especially related as there is no meta-learning of those adapters and instead scaling and shifting coefficients are learned to account for domain discrepancy, like in DRA.

Section A.1 (in the Appendix) states that a flexible finetuning algorithm that adapts test-time finetuning based on number of shots and ways has not been explored. In fact, though, it has been explored. [6]


## Address clarity issues (ordered in decreasing importance, roughly)

The authors claim that their approach follows the pretraining - meta-training - finetuning pipeline (e.g. in the introduction they say “In this paper, following the pretrain-meta-train-finetune pipeline [...]” but nowhere in the paper is a meta-training phase mentioned. Their entire model section is dedicated to the finetuning part. There could be a meta-training phase where episodes are used and the same finetuning procedure is applied in each episode, and perhaps some of the learnable parameters can be trained in this fashion, but the authors don’t mention anything to that effect. This was a serious clarity issue for me as I kept wondering what was meta-learned while reading the paper, and it’s still not clear to me (I assume there is no meta-learning).

From the experimental section, I understand that the authors re-trained DINO, to ensure it is only trained on the 712 classes of ImageNet that are in Meta-Datet’s meta-training set (which is great!). I would update the text accordingly, as phrases like “off-the-shelf” (e.g. in the Pre-training paragraph on page 5) make it sound like an already-trained one was used which would make for unfair comparisons.

What is “normalized cosine similarity” ? Cosine similarity already involves some normalization, is some additional normalization going on here?

In Equation 5, shouldn’t the upper limit of the inner sum (the one iterating over m) be P^2 instead of P since according to the notation used there should be P^2 patches?

In section 3.6, the authors state “[...] the model parameters, which include the adapter, the prefix together with the transformation module and the projector”. Not clear what is meant by “adapter” here. I’m assuming it’s referring to the learnable vectors of DRA, but should be more precise, as DRA had not been referred to as the “adapter” before. Also not clear what is the “transformation module”. Is this referring to module g? Should use consistent names to refer to modules throughout the paper.

The authors say in the text that Figure 4 shows an Omniglot episode, but the caption of Figure 4 says the episode is from Traffic Signs. Please correct.

In section 3.6, the authors state “we further standardize x’ with a centering statistics” - are the EMAs used for this calculated anew using the support set of each episode? Please clarify.

The approach described in section 3.3 should not be referred to as finetuning, since no parameters are finetuned. The backbone is frozen and the new transformation \phi is learned from scratch (not finetuned). The same is true of the proposed ‘task finetuning’ which again should not be called finetuning (the prefix parameters and g are trained from scratch). Task adaptation is a more general and appropriate term.


## Address minor soundness issues
“Given extremely small number of labelled images, the strategy of finetuning all parameters will be prone to overiffing the data”. Is there some empirical evidence that the authors had in mind here? If so, please include a citation (or otherwise weaken this claim / re-think the motivation for this). In practice it turns out that finetuning is powerful and less prone to overfitting than originally thought. The fact that naive finetuning is SotA on difficult benchmarks like [3] is a testament to this.

“DINO builds up supervision based on a self-distillation framework. [...] Accordingly, such a pre-trained model shall have good cluster property even among cross domain images” - why is this the case? I understand the motivation behind self-supervised learning producing rich features but I’m not aware of any guarantee for generalizing well to (all) new domains? Please add a citation or revise.

The authors don’t say how validation was done for hyperparameters. Table 4 shows the results of different hyperparameters on the *test* set, but how did the authors pick the hyperparameters with which they computed the main results for their model? This is particularly relevant and important to comment on for generalization to new datasets and transparency here is important.

## Additional ablations

In section 3.6, the authors state “we further standardize x’ with a centering statistics” - it would be nice to also have an ablation that removes this component.

Despite the thorough ablations in this paper, there is still a confounding factor between the proposed method and a lot of the previous baselines: pre-training is done in a self-supervised manner here using DINO whereas traditionally this is done supervisedly (with a few exceptions, e.g. CTX uses a mixture of both, jointly). It is unclear how big of a role this difference plays to the success of this method. Ideally, the authors would re-run some of the previous methods starting from a self-supervised trained backbone, even for resnets. But perhaps the easiest way to bridge the gap is to train ViT-t/-s in a *supervised* only way (this is different from their experiments where supervised finetuning was attempted). This will be a useful datapoint to report and can give more insights into comparisons with previous work.


Non-critical changes that would strengthen the paper
========================================

Though I really appreciate the authors re-training DINO on the meta-training set of ImageNet only for the main results, an additional experiment that shouldn’t be too hard to perform is using the pre-trained DINO off-the-shelf (trained on the entire ImageNet), in order to get apples-to-apples comparisons with P>M>F which is the most closely-related out of the methods in Table 2, since it also uses ViT-s. This would allow to better gauge the merit of the proposed adaptation approach in a more meaningful way.

In the set of experiments for the question: “Do we need a more complex adapter structure?” - another variant to explore here is learning both additive and multiplicative parameters (instead of additive ones only like the proposed method). This would be more akin to FiLM conditioning which learns an affine transformation.

The combined Meta-Dataset+VTAB benchmark [3] is arguably a more appropriate choice for evaluating this work. One reason for this is that the entire ImageNet is fair game for meta-training in that case, so different off-the-shelf models can actually be used. Further, it opens the door to comparisons against larger transfer learning models that were evaluated there, which is very relevant for this research direction.


Some typos and wording suggestions (there are more, please proof read the paper)
====================================================================
‘predominate’ → ‘predominant’ (in the introduction)

‘much less model parameters’ → ‘much fewer model parameters’ (in several places)

‘fineuning’ → ‘finetuning’ (in related work)


‘based data’ → ‘base data?

‘analogous information’ → ‘consistent information’ (in section 3.6)


References
==========
[1] Exploring the Limits of Large Scale Pre-training. Abnar et al.

[2] Head2Toe: Utilizing Intermediate Representations for Better Transfer Learning. Evci et al.

[3] Comparing Transfer and Meta Learning Approaches on a Unified Few-Shot Classification Benchmark. Dumoulin et al.

[4] Fast and Flexible Multi-Task Classification Using Conditional Neural Adaptive Processes. Requeima et al.

[5] Learning a Universal Template for Few-shot Dataset Generalization. Triantafillou et al.

[6] Learning to Balance: Bayesian Meta-Learning for Imbalanced and Out-of-distribution Tasks


**Strengths And Weaknesses:**

Strengths
========
[+] Exploring adaptation algorithms for few-shot learning with ViT backbones is interesting given the recent popularity of this architecture, and relatively underexplored

[+] The proposed model with ViT-small outperforms the ResNet34 variant approaches (ResNet34 has roughly the same number of parameters as ViT-small) on 7 out of 10 datasets

[+] The authors perform a thorough ablation study, and I also like the fact that they report sensitivity to hyperparameters (Table 4).

Weaknesses
==========
[-] While for the most part the paper is well-written, there are some clarity issues, one of which is particularly severe (see detailed comments below)

[-] Some related work is missing

[-] The proposed method does not improve upon previous works on all datasets. More research is needed to understand those limitations and tradeoffs.

---

> ### Author Response · Authors · 2022-08-01
> **Response to Reviewer hyDZ (Part 1)**
>
> Thank you for your constructive comments and suggestions. Our responses are as follows.
>
> 1. Add missing related work
>
> Thanks for the suggestion. We have cited these works along with the corresponding discussion in the revised version. Specifically, (1) [3, 4] mainly need meta-training to learn proper parameter generator while our eTT does not need meta-train, and (2) while [6] provides a flexible finetuning algorithm for FSL,  it is non-trivial to directly merge such methods with our proposed eTT due to different network structures and tuning strategies. It can be taken as the future work to properly utilize these flexible finetuning algorithms to further improve the performance of ViT in FSL.
>
> 2. Clarity issues
>
>  > a)   The authors claim that their approach follows the pretraining - meta-training - finetuning pipeline (e.g. in the introduction they say “In this paper, following the pretrain-meta-train-finetune pipeline [...]” but nowhere in the paper is a meta-training phase mentioned...
>
>  Thanks. We clarify that we do not have meta-training phase. The new parameters utilized in finetuning phase do not need meta-train for some specific initialization. Instead, the proposed attentive prototype is used to initialize APT and simple zero-initialization is used for DRA. We rephrase this point in the introduction.
>
>  > b)   ...I would update the text accordingly, as phrases like “off-the-shelf” (e.g. in the Pre-training paragraph on page 5) make it sound like an already-trained one was used which would make for unfair comparisons.
>
>  Thanks. We have clarified this point in the revised manuscript.
>
>  > c)   What is “normalized cosine similarity” ? Cosine similarity already involves some normalization, is some additional normalization going on here?
>
> Sorry, typos. We mean the “ normal cosine similarity”. It is the standard cosine similarity and no additional normalization is used. We have revised this term and added the corresponding equation form of cosine similarity.
>
>  > d)   In Equation 5, shouldn’t the upper limit of the inner sum (the one iterating over m) be P^2 instead of P since according to the notation used there should be P^2 patches?
>
>  Sorry typos, it should be $P^2$. We have revised accordingly.
>
>  > e)   ...Not clear what is meant by “adapter” here?what is the “transformation module”. Is this referring to module g?
>
> Thanks, “Adapter” denotes the proposed DRA here; and  “transformation module” means module $g$. Explanation has been added to the manuscript. We double check the paper and make sure to use consistent names to refer to the modules throughout the paper.
>
>  > f)   The authors say in the text that Figure 4 shows an Omniglot episode, but the caption of Figure 4 says the episode is from Traffic Signs. Please correct
>
> Sorry typos. We have corrected the text.
>
>  > g)   In section 3.6, the authors state “we further standardize x’ with a centering statistics” - are the EMAs used for this calculated anew using the support set of each episode? Please clarify.
>
> Yes, we clarify that our standardization strategy is the same as DINO. Specifically, for each episode we maintain an exponential moving average (EMA) of $\bar{x}'$ as the center variable $c_{center}$. Before calculating $\ell_{dist}$, we standardize $\bar{x}'$ as $\sigma(\frac{\bar{x}'-x_{center}}{\tau})$, where $\sigma$ denotes softmax function and $\tau$ is the temperature. We have added a detailed description in our paper.
>
> > h) The approach described in section 3.3 should not be referred to as finetuning...
>
> Thanks. We have revised the terms used in our paper.

---

> > ### Comment · Reviewer_hyDZ · 2022-08-14
> > **thank you for the responses!**
> >
> > Thank you for the thorough responses.
> >
> > I am confused by this point: "Specifically, (1) [3, 4] mainly need meta-training to learn proper parameter generator". This is true of [4] only out of the list of references I provided in my original review.
> >
> > The table in part 2 of the response convincingly makes the point of decreased performance when finetuning the parameters of the network using very small numbers of shots. I would still argue that this alone doesn't prove there is an *overfitting* effect (though I don't disagree that this is *probably* what's going on, based on intuition). To quantify overfitting, we need to look for symtpoms beyond just decreased performance (as that could have also been due to underfitting for instance), like the relationship between a train and validation set accuracy. Regardless, I agree with the general point made by the authors here and agree that eTT seems a more promising way to perform adaptation especially using very few shots.
> >
> > The additional ablations with different DINO variants and supervised pre-training are really interesting, thank you for the discussion. It's interesting that self-supervision meshes particularly well with ViT-like backbones and can yield nice gains combined with the proposed adaptation method.
> >
> > re: standardization: thank you for the additional ablations. Is it safe to conclude that these results without standardization are worse, though, as stated in the response? the numbers look pretty similar to me. What is the observation that these are worse based on? It would be good to at least show 95% confidence intervals.
> >
> > Thank you for all the clarifications, rewording updates made to the paper.

---

> > > ### Author Response · Authors · 2022-08-16
> > > **Thank you for your comments**
> > >
> > > 1. I am confused by this point: "Specifically, (1) [3, 4] mainly need meta-training to learn proper parameter generator". This is true of [4] only out of the list of references I provided in my original review.
> > >
> > > Sorry, there’s a typo: [3] was mistakenly added. We were meant to refer to [4] only.
> > >
> > >
> > > 2. The table in part 2 of the response convincingly makes the point of decreased performance when finetuning the parameters of the network using very small numbers of shots. I would still argue that this alone doesn't prove there is an overfitting effect.
> > >
> > > |     | ilsvrc | omniglot | aircraft | cub | dtd | quickdraw | fungi | vggflower | traffic | mscoco | avg   |
> > > |-----|--------|----------|----------|-----|-----|-----------|-------|-----------|---------|--------|-------|
> > > | eTT |    100 |    99.99 |      100 | 100 | 100 |       100 | 99.79 |       100 |     100 |  99.15 | 99.89 |
> > > | FT  |    100 |    99.87 |      100 | 100 | 100 |       100 | 96.95 |       100 |     100 | 95.20  | 99.20 |
> > >
> > >
> > > Yes, we agree that only test accuracies cannot reflect the existence of overfitting. However, as we have stated in our response, the finetuning error of both methods generally converges on test episodes. To further support this claim we report the training accuracy under the same setting as in the original response, where episodes with minimum shot number no larger than 2 are selected. We can find that on most of the datasets both methods receive 100% accuracy, which means these two methods can well learn the support data. We think the overfitting can be confirmed given high training accuracies and worse testing performance.
> > >
> > > 3. re: standardization: thank you for the additional ablations. Is it safe to conclude that these results without standardization are worse, though, as stated in the response? the numbers look pretty similar to me. What is the observation that these are worse based on? It would be good to at least show 95% confidence intervals.
> > >
> > > |           | ilsvrc | omniglot | aircraft | cub  | dtd  | quickdraw | fungi | vggflower | traffic | mscoco |
> > > |-----------|--------|----------|----------|------|------|-----------|-------|-----------|---------|--------|
> > > | w/o stand | 1.06   | 1.25     | 1.05     | 0.89 | 0.64 | 0.92      | 1.05  | 0.38      | 0.96    | 0.96   |
> > > | w stand   | 0.97   | 1.22     | 1.06     | 0.91 | 0.57 | 0.87      | 1.06  | 0.46      | 0.90    | 0.99   |
> > >
> > >
> > > We provide the confidence interval of these two variants in the above table, in which we can find the confidence intervals are similar among all variants on each dataset. We think it is safe enough to claim that the standardization strategy is effective, given two facts. First, the results with standardization are consistently better than those without standardization, which can hardly be resulted from random effects. Second, the performance gap is especially obvious to datasets like Omniglot and MSCOCO.
> > >
> > > 4. Thanks for the reviewer’s efforts on suggestions and comments.

---

> ### Author Response · Authors · 2022-08-01
> **Response to Reviewer hyDZ (Part 2)**
>
> 3. Minor soundness issues.
>
>  > a)   “Given extremely small number of labelled images, the strategy of finetuning all parameters will be prone to overiffing the data”. Is there some empirical evidence that the authors had in mind here? ...
>
>  |                     | ILSVRC | Omni  | Acraft | CUB   | DTD   | QDraw | Fungi | Flower | Traffic | COCO  | Avg   |
> |-----|--------|----------|----------|-------|-------|-----------|-------|-----------|---------|--------|-------|
> | eTT |  40.22 |    64.79 |    41.33 | 55.11 | 66.20 |     49.14 | 56.33 |     85.03 |   75.29 |  56.19 | 58.96 |
> | FT  |  29.19 |    54.54 |    35.10 | 41.54 | 53.66 |     43.37 | 38.53 |     76.76 |   72.90 |  41.21 | 48.68 |
>
> Thanks. Indeed [3] has shown that simple finetuning can generally have good results on Meta-Dataset. However, the setting in [3] is made up of fixed number of support images (1000) on VTAB and random number of support images (50-500) on Meta-Dataset. [3] only reports the average accuracies among all episodes, while not studying the case which support set only contains extremely few images, say only 1 or 2 samples per class. In contrast, by finetuing the machine learning models on such scarce labeled data will easily lead to overfitting, which is empirically shown by previous works including [i, ii, iii] (cited in the paper).
>
> Furthermore, to reveal if such a problem exists in Meta-Dataset, we conduct an experiment as follow: during a normal testing phase, we select all episodes whose minimum shot (minimum number of support images for each class) is no larger than 2 (extremely small number of labelled instances), and compare the average accuracies of eTT and simple finetuning based on these episodes. The results are shown in the above table, from which we can find that given the fact that training loss of both methods converges on all datasets, finetuning is much worse than eTT. We think this result can prove that overfitting does exist when trying to optimize the whole backbone model, and to some extent our proposed eTT can fix this problem.
> We rephrase this sentence to make it much clearer.
>
> [i] Snell J, Swersky K, Zemel R. Prototypical networks for few-shot learning[J]. Advances in neural information processing systems, 2017, 30.
>
> [ii] Fei-Fei L, Fergus R, Perona P. One-shot learning of object categories[J]. IEEE transactions on pattern analysis and machine intelligence, 2006, 28(4): 594-611.
>
> [iii] Lester B, Al-Rfou R, Constant N. The power of scale for parameter-efficient prompt tuning[J]. arXiv preprint arXiv:2104.08691, 2021.
>
>
>  > b)   “DINO builds up supervision based on a self-distillation framework. [...] Accordingly, such a pre-trained model shall have good cluster property even among cross domain images” - why is this the case?
>
>  The claim is an empirical one based on our experiments which show good results based on DINO-pretrained models. We have clarified this in our paper.
>
>  > c)   The authors don’t say how validation was done for hyperparameters.
>
> For simplicity, the selection of hyper-parameters is conducted on the meta-validation set of ImageNet, which is the only within-domain setting in Meta-Dataset. Based on this setting, the generally consistent optimality of the hyper-parameters over all datasets indicates the proposed method is generalizable enough for cross domain datasets. We have clarified this in our paper.

---

> ### Author Response · Authors · 2022-08-01
> **Response to Reviewer hyDZ (Part 3)**
>
> 4. Additional ablation
>
>  > a)   In section 3.6, the authors state “we further standardize x’ with a centering statistics” - it would be nice to also have an ablation that removes this component
>
>  |                     | ILSVRC | Omni  | Acraft | CUB   | DTD   | QDraw | Fungi | Flower | Traffic | COCO  | Avg   |
> |:-------------------:|--------|-------|--------|-------|-------|-------|-------|--------|---------|-------|-------|
> | w/o standardization |  66.72 | 74.20 |  78.42 | 85.06 | 87.01 | 70.34 | 61.64 |  96.51 |   84.23 | 61.08 | 76.52 |
> | w standardization   |  67.37 | 78.11 |  79.94 | 85.93 | 87.62 | 71.34 | 61.80 |  96.57 |   85.09 | 62.33 | 77.61 |
>
> We compare the models with and  without standardization which is depicted in Sec 3.6. The results are shown in the above table. When not using standardization, the results are generally worse. The results verify that the effectiveness of this strategy. We have added this experiment into our main paper.
>
>  > b)   Despite the thorough ablations in this paper, there is still a confounding factor between the proposed method and a lot of the previous baselines: pre-training is done in a self-supervised manner here using DINO whereas traditionally this is done supervisedly (with a few exceptions, e.g. CTX uses a mixture of both, jointly).
>
>  |                 | ILSVRC | Omni  | Acraft | CUB   | DTD   | QDraw | Fungi | Flower | Traffic | COCO  | Avg    |
> |:---------------:|--------|-------|--------|-------|-------|-------|-------|--------|---------|-------|--------|
> | TSA DINO        |  48.18 | 64.94 |  56.74 | 45.49 | 69.06 | 59.51 | 31.13 |  81.01 |   48.70 | 26.18 | 53.09 |
> | eTT Sup |  65.17 | 67.47 |  73.30 | 87.71 |  84.50 | 67.46 | 55.51 |  92.55 |   64.08 | 63.68 | 72.14 |
> | Proto DINO |  63.37 | 65.86 |  45.11 | 72.01 |  83.50 | 60.88 | 51.02 |  92.39 |   49.23 | 54.99 | 63.84 |
> | LT+NCC DINO |  65.96 | 67.62 |  64.03 | 77.10 |  83.46 | 63.88 | 57.79 |  93.13 |   66.91 | 56.04 | 69.59 |
> | eTT DINO      |  67.37 | 78.11 |  79.94 | 85.93 | 87.62 | 71.34 | 61.80 |  96.57 |   85.09 | 62.33 | 77.61 |
>
> As suggested, we conduct additional ablation study, including (1) TSA DINO: We follow the official implementation of DINO-ResNet50 to pretrain a ResNet34 using DINO on ImageNet meta-train split. (2) eTT Sup: We follow DeiT to train a ViT-s on ImageNet meta-train split using supervised learning. The results are shown in the above table, together with three variants (Proto DINO: use ProtoNet to test on novel data with DINO pretrained features; LT+NCC DINO: The basic test-time finetuning method in Sec.3.3; eTT DINO: our main model used in the paper) of our eTT using DINO pretrained ViT-s collected from our main paper. We have added this experiment into our supplementary.
>
> In general, we can find that (1) While DINO can improve our eTT by about 5% on average, TSA DINO is surprisingly worse than the official reported results using supervised ResNet34. This indicates that DINO as a pretrain strategy plays different roles for these two types of backbones. The results are in accordance with the some experimental results mentioned in the original DINO paper [Caron et. at., 2021], where ViT-s shows higher kNN accuracies than ResNet50 when using DINO pretrain(Tab.13, Sec.B in Supplementary.). We think a potential reason is that while DINO can improve the self-attention mechanism to help it better focus on the objects of interest, which is verified in DINO, such a property can be  hardly generalized to the ResNet backbone. We believe this is a future work for the DINO backbones in general.
>  (2) The performance gap between eTT Sup and eTT DINO is smaller than that between Proto DINO/LT+NCC DINO and eTT DINO. This means that compared with the DINO pretrain strategy, our proposed eTT has a much more significant contribution to the performance. (3) eTT Sup performs worse on most datasets than eTT DINO. We think the reason is that with self-supervised pretrain, the backbone can learn more generalisable knowledge beyond the label information, which results in better performance when facing novel data. On the other hand, eTT Sup is better on CUB and COCO, which may be attributed to some kind of overlap between classes in these two datasets and meta-train split of ImageNet.

---

> ### Author Response · Authors · 2022-08-01
> **Response to Reviewer hyDZ (Part 4)**
>
> 5. Non-critical changes
>
>  > a)   an additional experiment that shouldn’t be too hard to perform is using the pre-trained DINO off-the-shelf
>
>  |                                 | ILSVRC    | Omni  | Acraft    | CUB       | DTD       | QDraw | Fungi     | Flower    | Traffic | COCO      | Avg    |
> |:-------------------------------:|-----------|-------|-----------|-----------|-----------|-------|-----------|-----------|---------|-----------|--------|
> | DINO meta-train |     67.37 | 78.11 |     79.94 |     85.93 |     87.62 | 71.34 |     61.80 |     96.57 |   85.09 |     62.33 | 77.61 |
> | DINO full        | **74.76** | 78.73 | **80.10** | **86.99** | **87.72** | 71.56 | **61.95** | **96.66** |   85.83 | **64.25** | 78.82 |
> | P>M>F                           |     74.69 | 80.68 |     76.78 |     85.04 |     86.63 | 71.25 |     54.78 |     94.57 |   88.33 |     62.57 | 77.53 |
>
> We follow the reviewer's suggestion to test our eTT using off-the-shelf DINO ViT-s model. The results are shown as above. We can find that (1) For those datasets on which DINO meta-train performs better than P>M>F,, using full ImageNet to train DINO can bring further improvement. (2) With the help of more data, our eTT overpasses P>M>F, on ILSVRC and MSCOCO. (3) While more data does improve the results on Omniglot and TrafficSign, the final results are still worse than those of P>M>F. This has been analyzed in our paper. To sum up, given all these results, as a lighter model in that no meta-training phase is utilized and only few parameters are engaged in the test-time tuning, our method can still enjoy comparable performance with P>M>F when training on the same amount of data.
>
>  > b)   In the set of experiments for the question: “Do we need a more complex adapter structure?” - another variant to explore here is learning both additive and multiplicative parameters
>
>  |               | ILSVRC | Omni  | Acraft | CUB   | DTD   | QDraw | Fungi | Flower | Traffic | COCO  | Avg   |
> |:-------------:|--------|-------|--------|-------|-------|-------|-------|--------|---------|-------|-------|
> | FiLM-like DRA |  66.91 | 75.32 |  78.26 | 85.78 | 86.83 | 70.29 | 61.65 |  96.50 |   84.48 | 61.75 | 76.78 |
> | Offset DRA    |  67.37 | 78.11 |  79.94 | 85.93 | 87.62 | 71.34 | 61.80 |  96.57 |   85.09 | 62.33 | 77.61 |
>
>
> In the above table we show the results of FiLM-like DRA, in which we add a scaling vector for each adapter as in FiLM layer. The difference is that we still use the original way to directly tune the parameters on the novel support sets, instead of using another meta-trained module to generate the parameters. As shown in the table, FiLM-like DRA is worse than offset DRA by about 0.8% on average, while doubling the parameter size based on offset DRA. We have added this experiment into our paper.
>
>  > c)   The combined Meta-Dataset+VTAB benchmark [3] is arguably a more appropriate choice for evaluating this work.
>
>  Thank you for your suggestion. We agree that MD+VTAB is indeed a suitable benchmark for our model. However, given its extremely large scale, it is hard to conduct the experiments at once due to the difference between our code and the original Meta-Dataset. It really demands very expensive computations to run Meta-Dataset+VTAB benchmark. Besides, the current results on Meta-Dataset are enough to validate the effectiveness of our method. We will take the experiments on MD+VTAB as a future work.
>
>  6. Typos
>
>  Thank you. We have revised accordingly.
>
>  7. The proposed method does not improve upon previous works on all datasets. More research is needed to understand those limitations and tradeoffs.
>
>  We have analyzed the inferior performance on Omniglot in Sec. 4.2, along with the limitation of our method in Sec. A.1. Our paper is aimed to provide a new perspective of how ViTs can be efficiently and effectively used in FSL. The current results can basically support the efficacy of our method when comparing with ViT-based PMF or ResNet-based complex methods like CTX. It will be a future work to further improve the performance especially on datasets like Omniglot.

---

### Review · Reviewer_EfBc · 2022-07-13

**Summary Of Contributions:**

This paper proposes a transformer architecture for few-shot adaptation. They first train the transformer backbone on self-supervised data using DINO, and then fine-tune the architecture using the provided base-classes. However, instead of fine-tuning the entire network, they only place "prefixes" to each transformer layer before key and query modules and only fine-tune corresponding parameters, which are light-weight and small. They also design domain residual adapter and prototype regularization to improve training. Prefixes are initialized using class-conditioned embedding of the samples. Experiments show competitive results on Meta-dataset.

**Broader Impact Concerns:**

I do not see any significant ethical concerns specific to this work.

**Requested Changes:**

1. Add more analysis on the domain residual adapter.

2. If possible, comparisons with prior works also using DINO SSL.

3. Comparison with PMF under equal amounts of data.

**Strengths And Weaknesses:**

- Strengths

1. The idea of only fine-tuning the prefix parameters is interesting, and optimized to train.
2. The performance of the method overall is quite strong, and the ablations further provide usefulness of each module.
3. The problem setting and the proposed modules have been well-explained.

- Weaknesses

1. While the idea of attaching prefixes to the transformer has been explained and ablated well, the DRA module analysis is lacking. It is claimed that the DRA module is used to bridge the domain shifts between base and few shot classes which have considerable domain gap. This is not clear from the experiments Table 3, and it is not clear that the benefits of the proposed DRA are especially observable in cases of domain shift.

2. Pretraining on DINO is a choice that is not present in prior works, but only present in this work. So I am curious to know how much of performance improvement is due to this pretraining strategy? From Fig 3, LT+NCC also benefits heavily from DINO pre-training, so what would be the performance of methods in Table 2 like TSA with this pre-training?

3. It is also argued that PMF uses a lot more data, but can you also perform a like-to-like comparison by adding that data to your model as well? When presented with equal data, does your method do better or still worse than PMF?

4. In general, it is easier for readers to have an additional column in Tables 2 and 3 stating the average accuracy along with rank.

5. The improvements provided by prototypical initialization seems to be marginal. Rando already seems to be a competitive baseline, and avg is almost equal to the heuristic proposed on most datasets.

6. You mention that you set the number of prototypes is equal to N, but if I understand correctly, isn't N sampled uniformly from [5, N_max] as stated in sec 3.1?

---

> ### Author Response · Authors · 2022-08-01
> **Response to Reviewer EfBc (Part 1)**
>
> Thank you for your constructive comments and suggestions. Our responses are as follows.
>
> 1. > While the idea of attaching prefixes to the transformer has been explained and ablated well, the DRA module analysis is lacking. It is claimed that the DRA module is used to bridge the domain shifts between base and few shot classes which have considerable domain gap. This is not clear from the experiments Table 3, and it is not clear that the benefits of the proposed DRA are especially observable in cases of domain shift.
>
> Thanks. We give the experimental results of DRA module in Tab.3. Such results can reflect the efficacy of our DRA. We give more interpretations as follows. The idea of using residual adapter to handle domain gap stems from [a, b] that efficient parameterization, especially simple convolution layers, is sufficient for domain-specific differences given universal features from large pretraining tasks. We take such an inspiration to further propose a novel adapter module that is suitable for ViTs in FSL and contains much fewer parameters. As for the experiment results, Tab. 3 shows that APT generally has significant performance gap with full eTT on out-of-domain datasets especially Omniglot, Aircraft and TrafficSign, in contrast to the in-domain testing episodes on ILSVRC with only 0.62% improvement. This means APT cannot handle the domain gap between the meta-train set and these datasets, and DRA can help the model more on the out-of-domain datasets. Compared with that, directly adopting DRA without APT performs poorly on Omniglot, Aircraft and Fungi, which means the merit of DRA cannot be attributed to its capacity.
>
> We visualize the self-attention map from models with and without DRA on ILSVRC and TraffignSign in Fig.5 in the supplementary. Specifically, we randomly sample an episode from each dataset and use our eTT to tune the model based on the support samples. Then we calculate the self-attention map of the last layer's class token and highlight the areas with top 20\% attention scores. We can find that for the in-domain ILSVRC episode, the model can attend to similar regions no matter whether DRA is used. In contrast, the model without DRA can easily attend to background regions with less valuable information, which reveals a potential reason that these two models have similar accuracies on ILSVRC but large performance gap on TrafficSign.
>
> [a] Rebuffi S A, Bilen H, Vedaldi A. Efficient parametrization of multi-domain deep neural networks[C]//Proceedings of the IEEE Conference on Computer Vision and Pattern Recognition. 2018: 8119-8127.
>
> [b] Deecke L, Hospedales T, Bilen H. Latent domain learning with dynamic residual adapters[J]. arXiv preprint arXiv:2006.00996, 2020.

---

> ### Author Response · Authors · 2022-08-01
> **Response to Reviewer EfBc (Part 2)**
>
> 2. > Pretraining on DINO is a choice that is not present in prior works, but only present in this work. So I am curious to know how much of performance improvement is due to this pretraining strategy? From Fig 3, LT+NCC also benefits heavily from DINO pre-training, so what would be the performance of methods in Table 2 like TSA with this pre-training?
>
>
>  |                 | ILSVRC | Omni  | Acraft | CUB   | DTD   | QDraw | Fungi | Flower | Traffic | COCO  | Avg    |
> |:---------------:|--------|-------|--------|-------|-------|-------|-------|--------|---------|-------|--------|
> | TSA DINO        |  48.18 | 64.94 |  56.74 | 45.49 | 69.06 | 59.51 | 31.13 |  81.01 |   48.70 | 26.18 | 53.09 |
> | eTT Sup |  65.17 | 67.47 |  73.30 | 87.71 |  84.50 | 67.46 | 55.51 |  92.55 |   64.08 | 63.68 | 72.14 |
> | Proto DINO |  63.37 | 65.86 |  45.11 | 72.01 |  83.50 | 60.88 | 51.02 |  92.39 |   49.23 | 54.99 | 63.84 |
> | LT+NCC DINO |  65.96 | 67.62 |  64.03 | 77.10 |  83.46 | 63.88 | 57.79 |  93.13 |   66.91 | 56.04 | 69.59 |
> | eTT DINO      |  67.37 | 78.11 |  79.94 | 85.93 | 87.62 | 71.34 | 61.80 |  96.57 |   85.09 | 62.33 | 77.61 |
>
> Yes, we add this ablation study to give more insights of our method. Specifically, as suggested, we conduct additional ablation study, including (1) TSA DINO: We follow the official implementation of DINO-ResNet50 to pretrain a ResNet34 using DINO on ImageNet meta-train split. (2) eTT Sup: We follow DeiT to train a ViT-s on ImageNet meta-train split using supervised learning. The results are shown in the above table, together with three variants (Proto DINO: use ProtoNet to test on novel data with DINO pretrained features; LT+NCC DINO: The basic test-time finetuning method in Sec.3.3; eTT DINO: our main model used in the paper) of our eTT using DINO pretrained ViT-s collected from our main paper. We have added this experiment into our supplementary.
>
> In general, we can find that (1) While DINO can improve our eTT by about 5% on average, TSA DINO is surprisingly worse than the official reported results using supervised ResNet34. This indicates that DINO as a pretrain strategy plays different roles for these two types of backbones. The results are in accordance with some experimental results mentioned in the original DINO paper [Caron et. at., 2021], where ViT-s shows higher kNN accuracies than ResNet50 when using DINO pretrain(Tab.13, Sec.B in Supplementary.). We think a potential reason is that while DINO can improve the self-attention mechanism to help it better focus on the objects of interest, which is verified in DINO, such a property can be hardly generalized to the ResNet backbone. We believe this is a future work for the DINO backbones in general.
>  (2) The performance gap between eTT Sup and eTT DINO is smaller than that between Proto DINO/LT+NCC DINO and eTT DINO. This means that compared with the DINO pretrain strategy, our proposed eTT has a much more significant contribution to the performance. (3) eTT Sup performs worse on most datasets than eTT DINO. We think the reason is that with self-supervised pretrain, the backbone can learn more generalizable knowledge beyond the label information, which results in better performance when facing novel data. On the other hand, eTT Sup is better on CUB and COCO, which may be attributed to some kind of overlap between classes in these two datasets and meta-train split of ImageNet.

---

> ### Author Response · Authors · 2022-08-01
> **Response to Reviewer EfBc (Part 3)**
>
> 3. > It is also argued that PMF uses a lot more data, but can you also perform a like-to-like comparison by adding that data to your model as well?
>
> |                                 | ILSVRC    | Omni  | Acraft    | CUB       | DTD       | QDraw | Fungi     | Flower    | Traffic | COCO      | Avg    |
> |:-------------------------------:|-----------|-------|-----------|-----------|-----------|-------|-----------|-----------|---------|-----------|--------|
> | DINO meta-train |     67.37 | 78.11 |     79.94 |     85.93 |     87.62 | 71.34 |     61.80 |     96.57 |   85.09 |     62.33 | 77.61 |
> | DINO full        | **74.76** | 78.73 | **80.10** | **86.99** | **87.72** | 71.20 | **61.95** | **96.66** |   85.83 | **64.25** | 78.82 |
> | P>M>F                           |     74.69 | 80.68 |     76.78 |     85.04 |     86.63 | 71.25 |     54.78 |     94.57 |   88.33 |     62.57 | 77.53 |
>
> Sure, we can add this experiment. The results still support our contributions. Specifically, we follow the reviewer's suggestion to test our eTT using off-the-shelf DINO ViT-s model. The results are shown as above. We can find that (1) For those datasets on which DINO meta-train performs better than PMF, using full ImageNet to train DINO can bring further improvement. (2) With the help of more data, our eTT overpasses PMF on ILSVRC and MSCOCO. (3) While more data does improve the results on Omniglot and TrafficSign, the final results are still worse than those of P>M>F, which we think may be correlated with the limitations of our method as analyzed in our paper. Given all these results, as a lighter model in that no meta-training phase is utilized and only few parameters are engaged in the test-time tuning, our method can still enjoy comparable performance with P>M>F when training on the same amount of data.
>
> 4. > In general, it is easier for readers to have an additional column in Tables 2 and 3 stating the average accuracy along with rank.
>
> Thanks. We have added average accuracy to all experiment results in the main context. Our averaged results are still better than those of competitors in general.
>
> 5. > The improvements provided by prototypical initialization seems to be marginal. Rando already seems to be a competitive baseline, and avg is almost equal to the heuristic proposed on most datasets.
>
> Thanks. These points actually support our contributions in this paper. Particularly, high performance of random indicates the effectiveness of the prefix tuning strategy, which requires few new parameters. Based on that, a good initialization can help further improve the model. Simply averaging image patches can lead to worse results on Omniglot and Aircraft than using random initialization, which is due to the noisy information introduced by background patches. Compared with that, our proposed attentive prototype leads to overall improvement on all datasets, which we think can prove the efficacy of the proposed initialization strategy.
>
> 6. > You mention that you set the number of prototypes is equal to N, but if I understand correctly, isn't N sampled uniformly from [5, N_max] as stated in sec 3.1?
>
> Thanks. In fact, N is available automatically during testing according to the standard FSL protocols. So we have clarified this in the paper.

---

### Review · Reviewer_3uwk · 2022-07-20

**Summary Of Contributions:**

This paper proposed a finetuning method for FSL, namely eTT, with two key components APT and DRA. By using eTT, finetuned ViT achieved consistently good performances on meta datasets, compared to ResNet-18 and -34 models.

**Requested Changes:**

Need to solve the "Weaknesses", including the novelty, detail, and reference related issues.

**Strengths And Weaknesses:**

Strength:
1. This work proposed an APT module that includes a well-designed initialization strategy for FSL.
2. It proposed an effective DRA module with only two learnable offset vectors for each transformer layer, which is low-cost and efficient.
3. It proposed a novel prototypical regularization for prefix distillation.

Weaknesses:
1. The proposed APT is based on the PT (P-tuning), which can thus hardly be considered an innovation module. Please highlight only the novel part, i.e., the initialization of the prefix, and compare it to using other initialization strategies (including naive methods such as the random initialization that has been used).
2. The illustration in Figure 1b is not easy to understand, even with the corresponding texts given in the Introduction section.
3. The quantitative comparison in Table 2 is confusing, is it highlighting the higher performance? In the column of ILSVRC, the 74.69 in row P>M>F* is higher than 67.37 in row Ours, while the 67.37 is highlighted. Similar confusion cases appear in Table 3.
3. It is more intuitive if add an average performance comparison in the quantitative evaluation.
4. Other details: In eq. (4), the m =1, …, P^2, while it is denoted in the line above eq. (4) as {xxx}_{m=1}^P, P or P^2, please keep consistent.
5. For the references, as the paper's main contribution is a finetuning method for FSL, it is helpful to cite and compare it to these two works: Meta-Learning with Latent Embedding Optimization (LEO, ICLR 2019); and Meta Transfer Learning for Few-shot Learning (MTL, CVPR 2019) which is also based on learning adaptive (scaling and shifting) weights for few-shot tasks while fixing pre-trained parameters. These methods used ResNet-12 (the most popular arch). It would be helpful if the proposed method can show more effectiveness than this popular architecture.

---

> ### Author Response · Authors · 2022-08-01
> **Response to Reviewer 3uwk (Part 1)**
>
> Thank you for your constructive comments and suggestions. Our responses are as follows.
>
>
> 1. > The proposed APT is based on the PT (P-tuning), which can thus hardly be considered an innovation module. Please highlight only the novel part, i.e., the initialization of the prefix, and compare it to using other initialization strategies (including naive methods such as the random initialization that has been used).
>
> Thanks! We clarify that our APT is based on P-tuning, while we introduce several novel components to improve the basic component. As suggested, we had cited and discussed the related works to highlight our novel parts in the paper.  Specifically, we have illustrated in our Sec. 2 and Sec. 3.4 that our APT is related to the prefix tuning and prompt tuning used in the previous works, and we have highlighted the difference in our proposed initialization. Particularly, we explain that “Such a prompting idea from NLP is inherited  and repurposed to  finetune a learnable prefix for each novel episode in this paper. However, these works [Li&Liang, 2021; Lester et. al., 2021;, Houlsby et. al., 2021] initialize the prefix or prompt with word embeddings which is not available in our problem. Instead, we propose an attentive prototype with regularization initializing the visual prefix with object-centric embeddings.“
> Moreover, the contribution of our paper lies in the novel exploration of such a method in FSL. As for the comparison with other initialization strategies, we have considered random initialization and naive averaging (Tab. 3 of main paper), which are the most commonly used ones in the previous papers.
> This shows the efficacy of our method.
>
> 2. > The illustration in Figure 1b is not easy to understand, even with the corresponding texts given in the Introduction section.
>
> Thank you. We have revised the teaser figure to make it easier to understand.
>
> 3. >The quantitative comparison in Table 2 is confusing, is it highlighting the higher performance? In the column of ILSVRC, the 74.69 in row P>M>F* is higher than 67.37 in row Ours, while the 67.37 is highlighted. Similar confusion cases appear in Table 3.
>
> Thanks. Yes, Tab. 2 and Tab. 3 is highlighting higher performance under the same setting. We have corrected the highlighted terms in Tab. 3. And we clarify that our method and P>M>F* have utilized different training data in Tab.2. Particularly, we highlight and report that P>M>F* adopts the full training set of ImageNet to pretrain the DINO ViT-s, which is different from our eTT that only utilizes the meta-train split with much fewer training samples. This fact has already been stated in our Datasets part in Sec. 4.1, and the caption of Tab. 2. As suggested by the reviewer hyDZ, we conduct another ablation studies using the DINO model trained on a full training set of ImageNet. The results are in the following table. We can find that (1) For those datasets on which DINO meta-train performs better than PMF, using full ImageNet to train DINO can bring further improvement.  (2) Especially, with the help of more data, the accuracy of our model on ILSVRC, together with MSCOCO, which is concerned by the reviewer overpasses that of P>M>F. (3) While more data does improve the results on Omniglot and TrafficSign, the final results are still slightly worse than those of P>M>F. This has been analyzed in the paper. Given all these results, as a lighter model in that no meta-training phase is utilized and only few parameters are engaged in the test-time tuning, our method can still enjoy comparable performance with P>M>F when training on same amount of data.
>
> |                                 | ILSVRC    | Omni  | Acraft    | CUB       | DTD       | QDraw | Fungi     | Flower    | Traffic | COCO      | Avg    |
> |:-------------------------------:|-----------|-------|-----------|-----------|-----------|-------|-----------|-----------|---------|-----------|--------|
> | DINO meta-train |     67.37 | 78.11 |     79.94 |     85.93 |     87.62 | 71.34 |     61.80 |     96.57 |   85.09 |     62.33 | 77.61 |
> | DINO full        | **74.76** | 78.73 | **80.10** | **86.99** | **87.72** | 71.20 | **61.95** | **96.66** |   85.83 | **64.25** | 78.82 |
> | P>M>F                           |     74.69 | 80.68 |     76.78 |     85.04 |     86.63 | 71.25 |     54.78 |     94.57 |   88.33 |     62.57 | 77.53 |
>
> 4. > It is more intuitive if add an average performance comparison in the quantitative evaluation.
>
> Thank you. Good point, and we have added the corresponding results in our paper. Our averaged results are still better than those of competitors in general.

---

> ### Author Response · Authors · 2022-08-01
> **Response to Reviewer 3uwk (Part 2)**
>
> 5. > Other details: In eq. (4), the m =1, …, P^2, while it is denoted in the line above eq. (4) as {xxx}_{m=1}^P, P or P^2, please keep consistent.
>
> Thank you. We have revised this term to $P^2$.
>
> 6. > For the references, as the paper's main contribution is a finetuning method for FSL, it is helpful to cite and compare it to these two works: Meta-Learning with Latent Embedding Optimization (LEO, ICLR 2019); and Meta Transfer Learning for Few-shot Learning (MTL, CVPR 2019) which is also based on learning adaptive (scaling and shifting) weights for few-shot tasks while fixing pre-trained parameters. These methods used ResNet-12 (the most popular arch). It would be helpful if the proposed method can show more effectiveness than this popular architecture.
>
> Thank you for the suggestion. We have added these papers to our related works.   Note that our paper utilizes the large-scale and challenging Meta-Dataset. This dataset has two key merits: (1)
> such a large-scale dataset can effectively enable the training of ViT-based models, which typically are more data hungry. The meta-dataset is much larger than miniImageNet and tieredImageNet.   (2) The target subsets of Meta-Dataset are from various data domains. Such various cross-domain testing datasets give unique challenges to the few-shot learning algorithms in general.
> In contrast, LEO and MTL are tested on miniImageNet and tieredImageNet of which the training set is much smaller than that of Meta-Dataset, and the testing episodes only have in-domain setting. So the comparison is better to be conduct on Meta-Dataset. So the comparison is better to be conducted on Meta-Dataset. As for LEO, it is hard to directly modify it into our setting without detailed design. As for MTL, we use the FiLM-like DRA to approximate the MTL method, as FiLM layer has the similar adapter structure to MTL (utilizing both scaling and shift vector instead of the offset vector as used in our DRA). To make MTL more comparable to our eTT, we  still use the original way to directly tune the parameters on the novel support sets.  To sum up, we conduct another ablation study and try to compare the adapter in MTL against our DRA by apple-to-apple.
> The results are shown below; and note that for both methods, only adapter component is changed (i.e. FiLM-like DRA V.S. our Offset DRA). As shown in the table, FiLM-like DRA is worse than offset DRA by about 0.8% on average, while doubling the parameter size based on offset DRA. This shows efficacy of our adapter component.
>
> |               | ILSVRC | Omni  | Acraft | CUB   | DTD   | QDraw | Fungi | Flower | Traffic | COCO  | Avg   |
> |:-------------:|--------|-------|--------|-------|-------|-------|-------|--------|---------|-------|-------|
> | FiLM-like DRA |  66.91 | 75.32 |  78.26 | 85.78 | 86.83 | 70.29 | 61.65 |  96.50 |   84.48 | 61.75 | 76.78 |
> | Offset DRA    |  67.37 | 78.11 |  79.94 | 85.93 | 87.62 | 71.34 | 61.80 |  96.57 |   85.09 | 62.33 | 77.61 |
>
> 7, Can the method use ResNet-12 architecture?
>
> No, as we have illustrated in our paper, this paper is aimed to make better usage of ViTs in FSL. Therefore our proposed method, including the DINO pretraining strategy and the APT are all designed for ViT structure and are inappropriate for ResNet structure to some extent.

---

### Author Response · Authors · 2022-08-01
**Summary of changes**

We would like to thank the reviewers for their constructive comments and suggestions. We have extensively revised the manuscript in terms of clarity, contribution and analysis. The revised part has been highlighted with red color. Specifically,
1. We have clarified several points such as the selection of hyper-parameters, the training procedure of our method and the name of our modules.
2. We have corrected the typos and wrong notations in the original version.
3. We have updated the teaser figure for better understanding.
4. We have added reference and discussion of more related works.
5. As suggested by the reviewers, we have conducted a bunch of additional ablation studies to further support the efficacy of our proposed eTT, including: a) comparison between models with and without standardization in prototypical regularization, b) FiLM-like DRA structure, c) comparison with baseline methods using DINO pretrain strategy, and d) comparison with models trained on full ImageNet. These results are added to the main paper and the supplementary material.

---

### Decision · Action_Editors · 2022-08-26

**Recommendation:** Accept with minor revision

**Comment:**

The paper was overall positively received by reviewers and received two "Learning Accept" and one "Accept" recommendation after the author response. The authors addressed most smaller and larger concerns of the reviewers in the revised manuscript, including clarifications and additional ablations and results.

Overall I recommend Accept with minor revision as the paper clearly validates experimentally its claims:
* Proposes novel and efficient model for few-shot on Vision Transformer (ViT)
* Provides solid experimental evaluation and good performance for the proposed model
* Presents clear ablation study

## Required changes for minor revision
* Please make sure that all promised changes in the author response are incorporated in the final version.
* Fix claim in abstract: The author acknowledged missing citations and included them in the main part of the paper “The general pipeline of Pretrain-(Meta- train)-Finetune has been explored in few ViTs on FSL (Hu et al., 2022; Evci et al., 2022; Abnar et al., 2021), recently.”, but missed to fix the abstract “with the only exception (Hu et al., 2022).”, which is now in contradiction.
* The author refer to (Hu et al., 2022) as “preliminary study” (page 5), which seems misleading given it is [now] a CVPR 2022 paper [this might not have been clear to the authors at submission time], and it should also be cited as such. Furthermore, the authors should check if other arxiv papers are published papers.
* Please use consistent bolding in tables and note it in the table caption, e.g. clarify why the top of table 6 is not bolded in the caption.
* Table 6: It is confusing why the numbers for the line “DINO meta-train” and “VIT-s+DINO” have identical numbers, but different names in the model column. Please revise and/or clarify in the caption.
* I am missing the fix of reviewer EfBc’s comment “You mention that you set the number of prototypes is equal to N, but if I understand correctly, isn't N sampled uniformly from [5, N_max] as stated in sec 3.1?”, the revised manuscript still states “N is first uniformly sampled from [5, Nmax] … on training or testing stage”. Please revise/clarify.

---

> ### Author Response · Authors · 2022-09-01
> **Response to Action Editor**
>
> Thank you for your comments.
>
> 1. Please make sure that all promised changes in the author response are incorporated in the final version
>
> 	Thank you. We have revised the manuscript according to reviewers’ comments.
>
> 2. Fix claim in abstract: The author acknowledged missing citations and included them in the main part of the paper “The general pipeline of Pretrain-(Meta- train)-Finetune has been explored in few ViTs on FSL (Hu et al., 2022; Evci et al., 2022; Abnar et al., 2021), recently.”, but missed to fix the abstract “with the only exception (Hu et al., 2022).”, which is now in contradiction.
>
>     This part has been revised as “Vision Transformers (ViTs) have rarely been taken as the backbone to FSL with few trials~\citep{hu2022pushing, evci2022head2toe, abnar2021exploring focusing on naive finetuning of whole backbone or classification layer.”
>
> 3. The author refer to (Hu et al., 2022) as “preliminary study” (page 5), which seems misleading given it is [now] a CVPR 2022 paper [this might not have been clear to the authors at submission time], and it should also be cited as such. Furthermore, the authors should check if other arxiv papers are published papers.
>
>     Thank you. We have checked the citations and fixed the problems.
>
> 4. Please use consistent bolding in tables and note it in the table caption, e.g. clarify why the top of table 6 is not bolded in the caption.
>
> 	In the new version we have bolded the best performing items in all tables except Tab.8 and Tab.10 in the appendix, given no actual meaning of comparing the scale of training accuracies and confidence interval.
>
> 5. Table 6: It is confusing why the numbers for the line “DINO meta-train” and “VIT-s+DINO” have identical numbers, but different names in the model column. Please revise and/or clarify in the caption.
>
>     Tab.6 contains results of two ablation studies focusing on the usage of full ImageNet as training set and the usage of DINO as pretrain algorithm, respectively. And both ‘DINO meta-train’ and ‘VIT-s+DINO’ refer to our final model in the main paper, while highlighting the different elements among each experiment. In the revision we have split apart these two experiments into two tables, and highlighted these two rows with consistent color, and added explanation in the caption.
>
> 6. I am missing the fix of reviewer EfBc’s comment “You mention that you set the number of prototypes is equal to N, but if I understand correctly, isn't N sampled uniformly from [5, N_max] as stated in sec 3.1?”, the revised manuscript still states “N is first uniformly sampled from [5, Nmax] … on training or testing stage”. Please revise/clarify.
>
> 	In fact the number of ways is designed to be available in FSL protocols, since basically the support set is labeled and each selected category has at least one support sample. We have explained in the revised version as follows, “N is supposed to be accessible knowledge during both training and testing. In the most naive case, one can get N by directly counting the number of support classes.”